# Propagation Paths and Source Distributions of Resolved Gravity Waves in ECMWF-IFS analysis fields around the Southern Polar Night Jet

Cornelia Strube[1], Peter Preusse[1], Manfred Ern[1], and Martin Riese[1]

[1]Institut für Energie- und Klimaforschung – Stratosphäre (IEK–7), Forschungszentrum Jülich GmbH, 52425 Jülich, Germany

**Correspondence:** Cornelia Strube (c.strube@fz-juelich.de)

**Abstract.** In the southern winter polar stratosphere the distribution of gravity wave momentum flux in many state-of-the-art climate simulations is inconsistent with long-time satellite and superpressure balloon observations around $60\,°$S. Recent studies hint that a lateral shift between prominent gravity wave sources in the tropospheric mid-latitudes and the location where gravity wave activity is present in the stratosphere causes at least part of the discrepancy. This lateral shift cannot be represented by the column-based gravity wave drag parametrisations used in most general circulation models. However, recent high-resolution analysis and re-analysis products of the ECMWF-IFS show good agreement to observations and allow for a detailed investigation of resolved gravity waves, their sources and propagation paths.

In this paper, we identify resolved gravity waves in the ECMWF-IFS analyses for a case of high gravity wave activity in the lower stratosphere using small-volume sinusoidal fits to characterise these gravity waves. The 3D wave vector together with perturbation amplitudes, wave frequency and a fully described background atmosphere are then used to initialise the GROGRAT gravity wave ray-tracer and follow the gravity waves backwards from the stratosphere. Finally, we check for indication of source processes on the path of each ray and thus quantitatively attribute gravity waves to sources that are represented within the model.

We find that stratospheric gravity waves are indeed subject to far (>1000 km) lateral displacement from their sources, taking place already at low altitudes (<20 km). Various source processes can be linked to waves within stratospheric GW patterns, such as the orography equator-ward of $50\,°$S and non-orographic sources above the Southern Ocean. These findings may explain why superpressure balloons observe enhanced gravity wave momentum fluxes in the lower stratosphere over the Southern Ocean despite an apparent lack of sources at this latitude. Our results also support the need to improve gravity wave parametrisations to account for meridional propagation.

# 1 Introduction

Gravity waves convey energy and momentum from sources mainly in the troposphere into the middle atmosphere and thus accelerate the mean flow (Alexander et al., 2010). Accordingly, general circulation models were widely shown to lack realism in describing the mean state of the atmosphere if they did not take the effect of gravity waves into account (Lindzen and Holton, 1968; McLandress, 1998; Manzini and McFarlane, 1998; Kim et al., 2003; Orr et al., 2010). Despite many improvements in parameterised gravity waves, a systematic delay in the springtime breakdown of the southern hemispheric stratospheric polar vortex due to insufficient gravity wave drag, referred to as the "missing drag problem", still occurs and is arguably the most recent example for this issue (e.g. Butchart et al., 2011; McLandress et al., 2012).

There have been attempts to solve the missing drag problem by enhancing orographic drag in the existing parametrisations (Garcia et al., 2017) or artificially adding gravity wave momentum flux (GWMF) as it would be induced by subgrid-sized mountains from small islands (Alexander and Grimsdell, 2013). However, a comparison of the model results with global observations of mesoscale gravity waves indicates that these approaches do not compensate for the whole discrepancy and hence a major effect must be of a different nature.

Already the first global observations of gravity waves in southern Hemisphere winter by the Microwave Limb Sounder (MLS) (Wu and Waters, 1996) and the CRyogenic Infrared Spectrometers and Telescopes for the Atmosphere (CRISTA) (Preusse, 2001; Ern et al., 2004) showed that the high wind velocities around the southern polar vortex were associated with an almost uniform band of enhanced gravity wave activity. The correlation of enhanced gravity wave activity in the middle atmosphere to the wind speeds was evident early on (Preusse et al., 2003). From the Southern Andes and the Antarctic Peninsula waves propagate to about $60°$S into the Drake passage, a mechanism which was further studied by a dedicated air plane campaign (Rapp et al., 2020). For gravity wave activity far off land, however, the main sources remained unclear. Success in reproducing the global distributions by uniform sources (e.g. Ern et al., 2006; Preusse et al., 2009a) does not resolve this puzzle and is rather misleading in suggesting local sources. Based on MLS measurements and model data from the European Centre for Medium-Range Weather Forecasts (ECMWF), Wu and Eckermann (2008) proposed first that the gravity waves observed in the polar vortex may be generated by the storm track regions at lower latitudes and propagate obliquely into the vortex. Recent observations by AIRS support these finding and show a southward component of the wave vector in the northern part of the jet and a northward component closer to Antarctica, thus indicating propagation into the jet core (Hindley et al., 2019).

Oblique propagation is a fundamental property of gravity waves. Considering the dispersion relation and group velocity of gravity waves it becomes evident that in an intrinsic frame of reference the group propagation of gravity waves occurs along their phase lines (Andrews et al., 1987). Given that most gravity waves have much longer horizontal than vertical wavelengths, oblique propagation has to be expected to be the regular case. For many waves then the ratio of horizontal and vertical wavelengths is comparable to the ratio of vertical to horizontal grid spacing in general circulation models, so that implementing vertical propagation but neglecting horizontal propagation across grid cells is a strong simplification. The exception for which lateral propagation is much less important are those mountain waves that are excited when the wind flows perpendicular over the ridge and the intrinsic horizontal group velocity compensates the advection of the wave packet with the

wind. In rare cases, oblique propagation can be directly observed. For instance, Sato et al. (2003) showed oblique propagation of a single wave packet in a case study based on radiosonde observations taken from a research vessel. In order to study propagation effects in more detail, Marks and Eckermann (1995) developed the Global or Regional Gravity Wave Ray Tracer (GROGRAT) based on the gravity wave ray-tracing equations formulated by Lighthill (1978). Using ray-tracers with a global launch distribution, Sato et al. (2009); Preusse et al. (2009a) and Kalisch et al. (2014) investigated the importance of oblique propagation for the gravity wave distributions in the upper stratosphere and mesosphere. Such modelling results are strongly supported by the patterns revealed from sub-annual cycle variations in a long time series of GWMF inferred from temperature measurements of the Sounding of the Atmosphere using Broadband Emission Radiometry (SABER) instrument (Chen et al., 2019). All these investigations point to a continuous mostly pole-ward propagation into the jet core and focusing of gravity waves in the mid stratosphere and higher up.

The concept of sources in the storm track is supported by an investigation of a global model run with $7\,km$ grid spacing presented by Holt et al. (2017). The authors show convection and frontogenesis around 40 to $50°\,S$ and corresponding maxima of GWMF at $15\,km$ altitude. However, already at approximately 18 to $20\,km$ altitude superpressure balloon observations show strong GWMF around $60°\,S$ at the lower edge of the stratospheric polar jet (Hertzog et al., 2008; Geller et al., 2013). Also observations from the High Resolution Dynamic Limb Sounder (HIRDLS) instrument as low as $20\,km$ altitude (Geller et al., 2013; Ern et al., 2018) show that the maximum is located further south than the storm tracks (around $60°\,S$). In contrast, above the storm tracks, where the sources are expected to be located, GWMF is not enhanced in these observations. Could the momentum flux from lower latitude sources be conveyed into the polar jet already in the lowermost stratosphere? Extreme oblique propagation in the upper troposphere and lower stratosphere (UTLS) region from observations of the Gimballed Limb Observer for Radiance Imaging of the Atmosphere (GLORIA) instrument over Iceland demonstrated by Krisch et al. (2017), as well as far propagation of gravity waves observed in radiosondes from Antarctica (Yoo et al., 2020), point to far oblique propagation at low altitudes being a candidate process. Could similar propagation pathways also explain the high GWMF in the southern hemisphere (SH) polar winter vortex?

We base our investigation on the results of a study from Ehard et al. (2017) who found indication for horizontal refraction and propagation of mountain waves from New Zealand into the SH polar vortex. The study highlights lidar observations from Lauder on 31 July and 1 August 2014, which showed a sudden decrease of wave amplitudes around $40\,km$ altitude. Simultaneously, AIRS observations featured a large-amplitude event of more than 2 million $km^2$ extent spanning from the South Island of New Zealand southeastward onto the ocean. Over the ocean, this wave field in AIRS measurements also spans far into the lower stratosphere. The gravity waves investigated in Ehard et al. (2017) remain over New Zealand up to $40\,km$ altitude and hence would not contribute to the lower stratospheric gravity wave field over the ocean. Thus, the waves found in AIRS measurements in the lower stratosphere cannot be the same gravity waves observed by the lidar and we are interested from where those waves originated from.

To search for the source of the lower-stratospheric gravity wave field, we use high-resolution, three-dimensional model data of the area under investigation. Modern high-resolution global models start to resolve more and more of the relevant part of the gravity wave spectrum. Comparison of observations with high-resolution model simulations show good overall agreement

in the distribution of GWMFes (e.g. Schroeder et al., 2009; Kim et al., 2009; Plougonven et al., 2013; Jewtoukoff et al., 2015). First attempts to rely only on resolved waves and not to use gravity wave parametrisations at all were made with the SKYHI general circulation model (Hamilton et al., 1999; Koshyk and Hamilton, 2001). Allowing shortest horizontal wavelengths of about 200 km, the Kanto model was able to produce a realistic middle atmosphere without parameterised gravity waves (Watanabe et al., 2008; Sato et al., 2009). Siskind (2014) have found that at a spatial sampling of 0.375°(T479) the general structure of the atmosphere is reasonably well represented, but that remaining biases still call for the need of a gravity wave parametrisation. The fact that even the higher spatial resolutions achieved by the general circulation model of ECMWF require a gravity wave parametrisation (Orr et al., 2010) may be due to the poor vertical resolution in the upper stratosphere and mesosphere (cf. Hamilton et al. (1999)) combined with a strong damping of gravity waves by the sponge layer of the model above 40 km altitude (Schroeder et al., 2009; Ehard et al., 2018). Recent very high horizontal resolution global simulations (1-3 km grid spacing) have the potential to resolve all gravity waves which could propagate freely into the stratosphere (Stephan et al., 2019a,b, and references therein) although these simulations are currently limited to a few-weeks simulation runs. It should be emphasised, however, that even at these high resolutions validation is required, and that GWMF could be even highly overestimated (Lane and Knievel, 2005).

These studies indicate that one may also investigate single gravity wave events in high resolution global model data, provided the synoptic scale wind and temperature structures are well represented as is the case for numerical weather prediction fields. In particular, the gravity wave wave field investigated in Ehard et al. (2017) shows up in ECMWF temperature analyses data. This offers the opportunity to fully characterise the wave field in terms of 3D structures by inferring wave vectors and wave amplitudes over the whole area. In our study, we use the full 3D characterisation to investigate the origin of this wave field based on ECMWF data. We show a case study of ray-tracing from stratospheric gravity waves analysed at 25 km altitude backwards in time and space to find the main pathway of the waves. Along the ray paths, we examine the model fields for likely gravity wave sources. Our aim is to differentiate in a quantitative way between local and remote sources and to assess the influence of lateral propagation and horizontal refraction. This case study can therefore provide important information on effects missing in current gravity wave parametrisations (Plougonven et al., 2020) and give some guidance of how these parametrisations could be improved.

The paper is organised as follows. The data and analysis methods are presented in section 2, the gravity wave structures and their origin are discussed in section 3, and the findings are put into context in Sect. 4 and summarised in the conclusions.

## 2  Data and analysis methods

### 2.1  ECMWF-IFS operational analyses

High resolution data are taken from the European Centre for Medium-Range Weather Forecasts (ECMWF) Integrated Forecast System (IFS). The ECMWF-IFS couples a general circulation model with a 4D-variational assimilation system. The assimilation of a wide range of in situ, ground-based and satellite data ensures a good representation of the current state of the atmosphere as basis for the predictions. The assimilation system thus confines the synoptic scale wind and temperatures. Grav-

ity waves, on the other hand, are considered as "atmospheric noise" and hence not assimilated, but generated self-consistently from the model. The spatial resolution of the ECMWF general circulation model has been continuously increased. For 2014, when the wave event considered in our work occurred, data are from IFS model cycle Cy38r2 which has a horizontal resolution of N640/T1279, corresponding to $\sim$16 km grid spacing (on the reduced Gaussian grid). Due to hyperdiffusion, the shortest horizontal wavelengths properly resolved are about 6-8 times this value, i.e. $\sim$100 km (Skamarock, 2004; Preusse et al., 2014), but in case of strong mountain wave forcing also shorter wavelengths may be contained. In the vertical, altitudes up to 80 km are represented by 137 vertical levels with a stepwise decreasing resolution starting at very fine sampling in the lower troposphere, around 250 m at the tropopause, about 2 km at the stratopause and even sparser sampling above.

An overview of a number of studies comparing ECMWF-resolved gravity waves with ground-based and satellite observations is given by Preusse et al. (2014): The ECMWF gravity wave temperature amplitudes seem to be in general underestimated by a factor of 1.5-2 at best. Inside this limit, gravity waves compare favourable up to about 40 km altitude both in global distribution as well as in wavelengths. Also phases often agree well with the results of observations. Observed and modelled gravity waves agree particularly well for orographic waves and waves in the jets, which likely are generated by spontaneous imbalance, but less well for gravity waves from convection (Schroeder et al., 2009). Above 40 km altitude, amplitudes are rapidly decreasing in ECMWF data due to the sponge layer of the model (Schroeder et al., 2009; Ehard et al., 2018).

For our study we use data interpolated to a constant longitude-latitude grid of $0.2° \times 0.2°$ and to geometric altitudes with a sampling of 500 m. This corresponds to the model vertical resolution at 25 km altitude, where we perform our wave analyses. The resolution is also adequate for the atmospheric background needed for performing ray-tracing studies with the GROGRAT gravity wave ray-tracer.

## 2.2 Meteorological situation

The wave event described in this paper was discovered in the frame of the Deep Propagating Gravity Wave Experiment (DEEP-WAVE) campaign (Fritts et al., 2016). The prevalent meteorological situations occurring during the campaign period is described in detail by Gisinger et al. (2017). Ehard et al. (2017), furthermore, explain the meteorological situation for 31 July and 1 August 2014 concentrating mainly at New Zealand's Southern Island. On the 31 July and 1 August 2014 this resulted in a northwesterly incident flow perpendicular to the Southern Alps mountain ridges in the troposphere.

Figure 1 presents the flow conditions in the upper troposphere and lower stratosphere at selected altitudes and times around the event. Figures 1(a) and (b) show the time evolution of the tropospheric jet from 31 July 2014 at 12:00 UTC, to 1 August 2014 at 12 UTC. The strongly meandering jet, for which the troughs and crests extend from $30°$S to $60°$S, is caused by synoptic-scale Rossby waves displacing the jet core. Between 31 July and 1 August, the jet core is moving above the southern tip of the South Island which leads to increased wind speeds over the Southern Alps and along the jet core in a streak southeast of the island. In the lower stratosphere, the wind turned by about $45°$ to westerly. At the altitude of 16 km (cf. Fig.1(c)), high horizontal wind speeds are especially present in a large area from the south tip of New Zealand down to $60°$S and extending east to about $170°$W about 12 h before the main event was observed at 1 August 2014 at 12:00 UTC. These conditions favour lateral gravity wave propagation at these altitudes. In the mid stratosphere, the flow was also westerly and the jet core farther

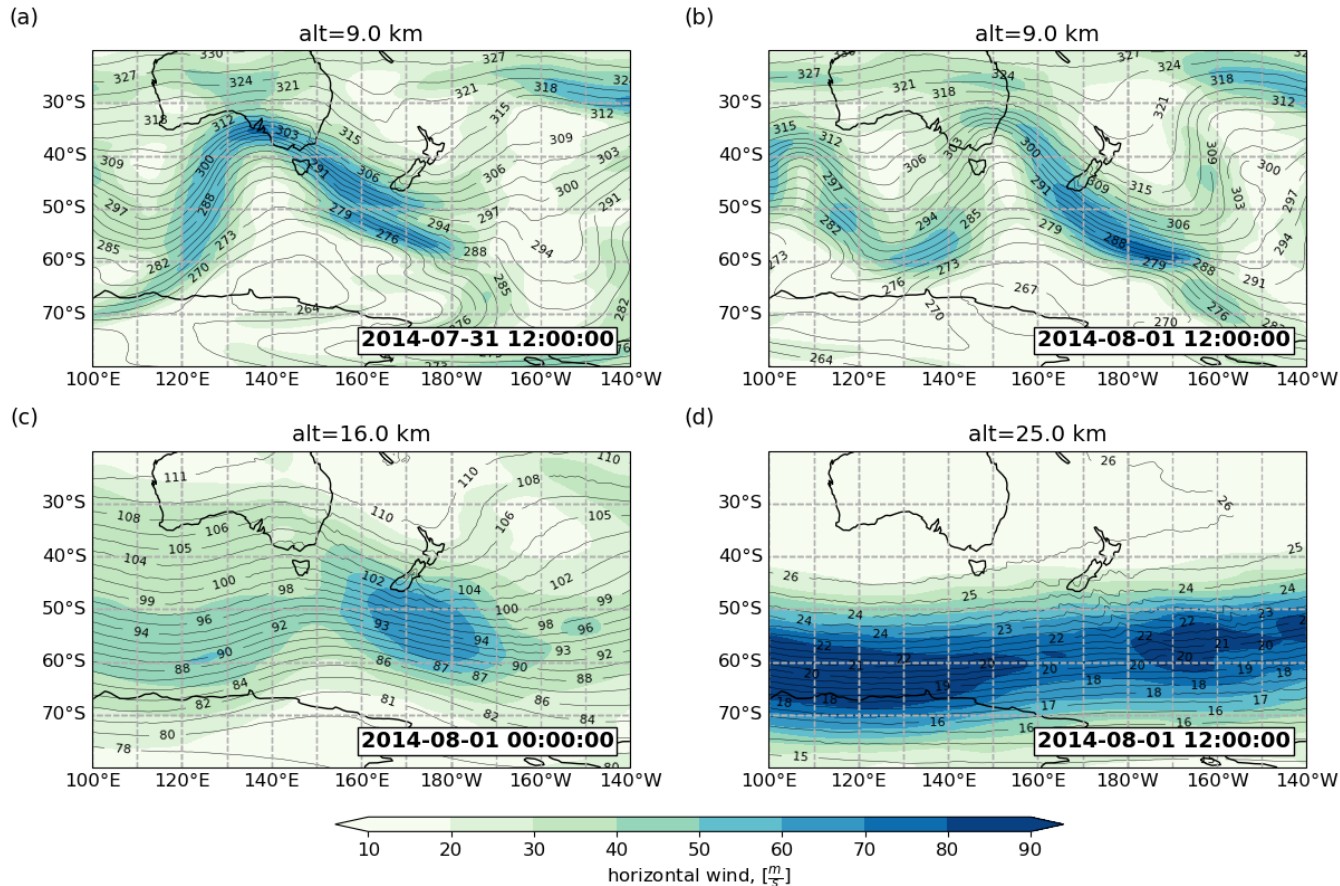

**Figure 1.** Tropospheric and stratospheric jet streams illustrated in horizontal wind speeds at different altitudes and times. The colour code shows horizontal wind speeds in $\frac{m}{s}$ overlaid by contours of the pressure in hPa from ECMWF-IFS analyses. The panels show (a) 9 km altitude at 31 July 2014, 12:00 UTC, (b) 9 km altitude at 1 August 2014, 12:00 UTC, (c) 16 km altitude at 1 August 2014, 00:00 UTC, and (d) 25 km altitude at 1 August 2014, 12:00 UTC.

south at 50 °S to 75 °S. The pressure contours in Fig. 1(d) show clear ripples in the area southeast of New Zealand. This already indicates the gravity wave activity, that we will discuss in more detail in this study.

### 2.3 Scale separation into background and fluctuations

To identify the signatures of gravity waves in ECMWF-IFS temperatures, we separate the large-scale background temperature $\bar{T}$ from small- and mesoscale temperature perturbations $\hat{T}$.

$$T = \bar{T} + \hat{T} \tag{1}$$

The background $\bar{T}$ is associated with the zonal mean state and dynamics of larger scales, like Rossby waves. The small- and mesoscale perturbations $\hat{T}$ are then associated with gravity waves. This separation of the gravity wave perturbations from a general background atmosphere is the first essential step of any gravity wave analysis from measurements or general circulation model data (Fetzer and Gille, 1994; Preusse et al., 2002; Baumgarten et al., 2017; Ehard et al., 2015; Ern et al., 2018). In particular, a horizontal scale separation is an effective mean to isolate gravity wave fluctuations, which has been shown recently by Strube et al. (2020).

We apply zonal spectral filtering to remove the background atmosphere which utilises the periodic nature of large-scale waves along a latitude circle. The zonal wavenumber spectrum is calculated using the Fast Fourier transform (FFT) for each height and latitude of a model snapshot. A cut-off wavenumber of 18 was applied in the low-pass filter to characterise the background. A zonal wavenumber 6 to 8 should in principle have sufficed to describe the Rossby waves for the targeted analysis altitude of 25 km and hence to isolate the gravity wave perturbations (Strube et al., 2020). However, we use the same scale separation to define the background atmosphere of temperature, winds and pressure from 0 to 45 km altitude for the ray tracing described below. A common scale separation is then necessary for the sake of uniformity of the background atmosphere. In particular in the tropopause region, higher wavenumbers are required to capture synoptic-scale structures. In addition to zonal wavenumber filtering, the low-passed spectral components are smoothed with a Savitzky-Golay filter applying a third-order polynomial over 5°latitude in the meridional direction and a fourth-order polynomial over 5.5 km in the vertical direction. Inverse FFT then retrieves the spatial temperature field representing the large-scale, smoothed background (correspondingly the wind and pressure fields for the full background atmosphere for ray tracing). The gravity wave associated temperature perturbations are defined by subtracting the background from the original temperature field.

Figure 2 shows horizontal and vertical cross sections through the gravity wave field found on 1 August 2014, 12:00 UTC, in the ECMWF-IFS temperatures. Figures 2a, 2b and 2c show temperature perturbations at altitudes of 25km, 12km and 40km, respectively, inferred with the smoothed zonal spectral filtering approach for scale separation described above. At all altitudes, a gravity wave field is stretching from the southern tip of New Zealand's southern island onto the ocean southeast of New Zealand.

At 25 km (Fig.2a) there is a large field of enhanced gravity wave amplitudes spanning from 170°E to 150°W and from 40°S to 65°S. This is the altitude selected for gravity wave characterisation in this study as will be described in Sect. 3.1. The gravity wave field is rather isolated and there are only few structures to the north and south as well as upstream of 160°E, especially for latitudes south of 50°S. Above New Zealand's South Island, wave structures with relatively short horizontal wavelengths are visible. Most pronounced is a short-wavelengths, high-amplitude patch in the South of the South Island with north-south alignment of the phase fronts at 25km altitude. Also weaker structures above the northern part of the South Island align mostly north-south in the phase fronts, which run parallel to the coast. This orientation was attributed to turning of the phase-fronts by horizontal refraction in Ehard et al. (2017). A resulting balance of the likewise turning background winds keeps the waves above the South Island of New Zealand, particularly observed for the region of Lauder, New Zealand.

To the south, we find a wave field with phase-fronts, generally directed from north-west to south-east which extend from the south tip of the South Island of New Zealand to about 63°S and 160°W. Closer inspection reveals that the phase fronts

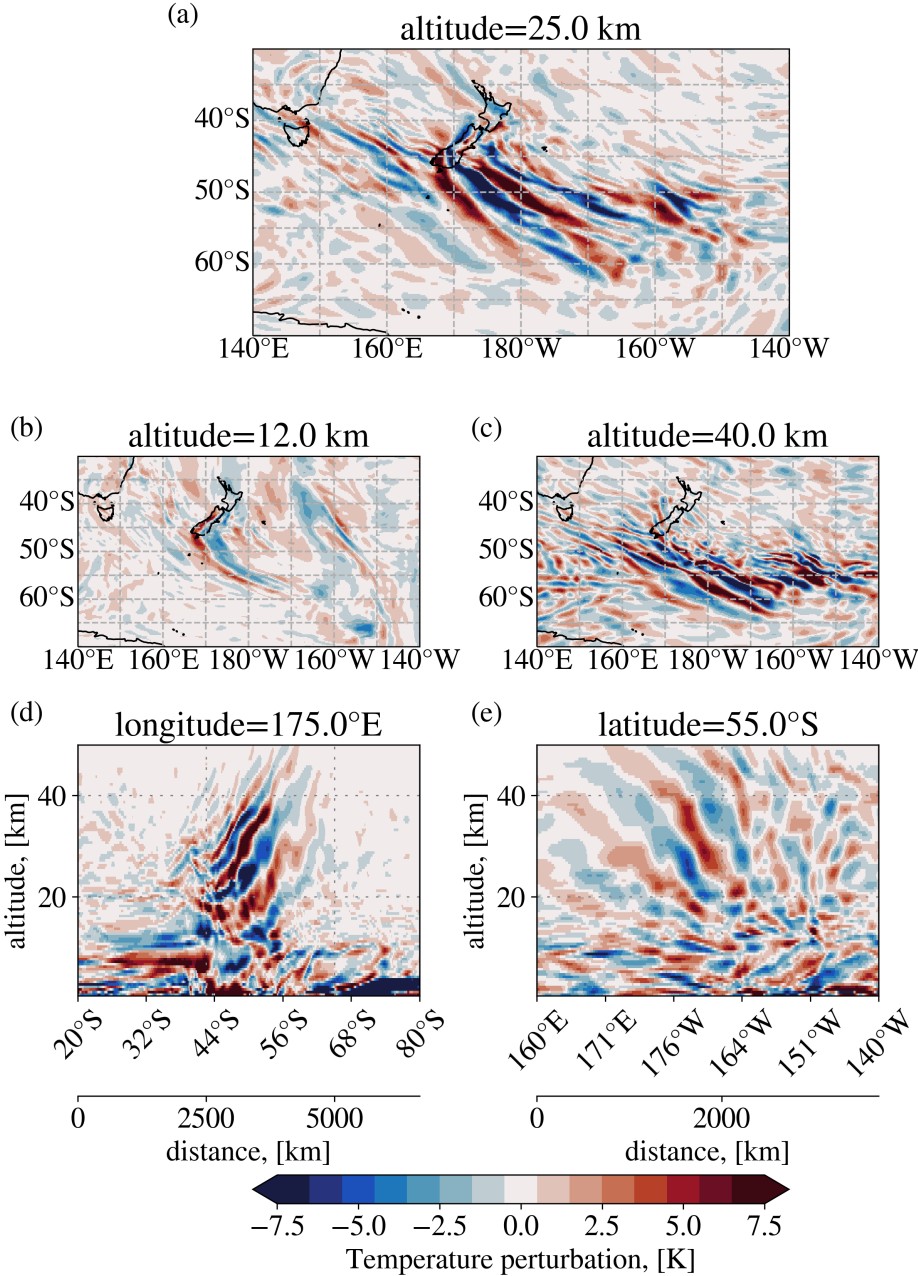

**Figure 2.** Horizontal (a-c) and vertical (d,e) cross sections of temperature perturbations found in the ECMWF-IFS temperature field on 1 August 2014, 12:00UTC. Horizontal cross sections are at (a) z=25, (b) z=12 and (c) z=40km altitude. Vertical cross sections show cuts along the meridian of (d) 175°E and the latitude circle of (e) 55°S. The temperature perturbations in the vertical cross sections are scaled with a factor of $\exp\left(-\frac{z-z_0}{2H}\right)$ with a reference altitude $z_0 = 25\,\mathrm{km}$ assuming a scale height of $H$=7 km to highlight the vertical phase line structure.

close to New Zealand (down to about 53°S) point to the southern part of the South Island. This is an indication of mountain waves. Mountain waves in favourable wind conditions propagate horizontally along the direction of the horizontal phase fronts (cf. Appendix A, Figure A1) and for mountain waves a spatial extension of the phase fronts, conversely, points to the source of the wave packet. Here, this source would be the southern part of the South Island. We will come back to this hypothesis in Sect. 3.3. South of 50°S the wave field is still coherent, but not homogeneous in terms of the wave properties. Instead, there are different horizontal scales and also some variation of the phase front direction is visible.

Below, at 12 km altitude the gravity wave field close to New Zealand is dominated by a V-shaped pattern of waves parallel to the mountain ridge line along the whole South Island and waves starting from the south tip of the South Island that extend in south-east direction to ∼50° S. The latter show similarity to the events analyzed by Jiang et al. (2019), and hence might suggest a trailing wave situation. The phase fronts of these waves point to the South Island and, therefore, indicate orographic origin, as discussed for waves at 25 km altitude. This wave structure seems to be continued for latitudes 51 °S to 58 °S, but with a different direction and the spatial extension of the phase fronts does not point to the island any longer. In longitudinal direction the wave pattern is restricted close to the island (extending from approximately 170 °E to 170 °W). At 12 km altitude, it is not trivial to distinguish between gravity wave signals and a potentially incomplete removal of synoptic-scale structures from weather systems (Strube et al., 2020). Therefore, it is difficult to interpret structures such as the ones seen at [60 °S, 160 °W] and [45°S, 160°W]. Also, there are some weaker patterns in a diagonal stripe from New Zealand upstream to the east coast of Australia.

At 40 km altitude (Figure 2c) the wave field has separated from the South Island of New Zealand. Some waves are also found upstream of New Zealand, and the gravity wave field reaches downstream to 150 °W, spanning thus in total more than 70 °of longitude. The phase fronts are directed rather to the west-east, indicating an additional refraction of the wave vector with wave directions turned further southward.

At 25 km, the most prominent horizontal wavelengths in the temperature residual fields appear to range between $\approx$300 km (e.g. seen around 55°S between 170°E and 180°E at 12 km, or south of 50°S between 160°W and 150°W at 40 km) and more than 1000 km (cf. 45°S to 50°S between 170°W and 180°W at 12 km, 55°S to 60°S between 170°E and 170°W at 25 km).

The amplitudes of the temperature perturbations are increasing with altitude. Maximum values range between about 7 K at 12 km and about 15 K at 40 km. Increasing amplitudes are expected from gravity wave theory if wave action is conserved, because the atmosphere has lower density at higher altitudes.

The temperature perturbations of the vertical cross sections in Figures 2d and 2e are scaled with a non-dimensional factor of $\exp\left(-\frac{z-z_0}{2H}\right)$, where $z$ is the altitude, $z_0$ a reference altitude of 25 km, and $H$ a density scale height estimate of 7 km. The scaling compensates for the density differences in altitude and hence emphasises the wave front structure. The two vertical sections cut through the center of the wave field: panel 2d gives a north-south section at 175°E, and panel 2e shows a west-east section at 55°S. The phase lines here are slanted to the west with altitude. This indicates waves propagating against the dominating eastward wind in this area. In the lower to mid stratosphere (between 20 and 40 km), the phase lines get steeper at higher altitudes, consistent with increasing wind velocity; above 40 km the phases are flattening again. Overall, the vertical

wavelengths range from about 5 km (cf. 45°S, between 160°E and 164°E at 35 km or around 165°W between 20 and 25 km) to about 20km (cf. 55°S, around 170°W above 25 km).

## 3 Results

### 3.1 Gravity wave characterisation

For the interpretation of the gravity wave structures, we apply the small-volume few-wave decomposition method S3D on a snapshot of ECMWF-IFS operational analysis temperature perturbations. The method was introduced by Lehmann et al. (2012) and previously used for gravity wave characterisation by Preusse et al. (2014), Krisch et al. (2017), Ern et al. (2017), Stephan et al. (2019a) and Stephan et al. (2019b). The considered region (the area between $40\,°$S and $70\,°$S in latitude and between $165\,°$E and $145\,°$W in longitude) is covered with small, overlapping analysis volumes (fitting cubes) for a systematic analysis. The volumes are defined by a number of grid points in each spatial direction on a longitude-latitude-altitude grid, which makes each volume quasi-rectangular for the small volumes. In each volume, the wave structure in the temperature perturbation is approximated by a monochromatic wave which is defined by

$$f(\boldsymbol{x}, \boldsymbol{k}) = A \sin(\boldsymbol{x}\boldsymbol{k}) + B \cos(\boldsymbol{x}\boldsymbol{k}) \tag{2}$$

with $\boldsymbol{x} := (x,y,z)$ the coordinate vector in zonal, meridional, and vertical direction and $\boldsymbol{k} := (k,l,m)$ the wave vector of zonal, meridional, and vertical wavenumber, respectively. The parameters $A$ and $B$ represent amplitudes.

The fit of the wave parameters is carried out by minimising the cost function,

$$\chi^2 = \sum_i \left( \boldsymbol{T'}_i - \sum_j f(\boldsymbol{x_i}, \boldsymbol{k_j}) \right)^2, \tag{3}$$

where $\boldsymbol{T'}$ represents the ECMWF-IFS temperature perturbations at each location $\boldsymbol{x}$. The amplitudes $A$ and $B$ can be calculated analytically after the wave vector is determined using standard methods for a least-squares fit. From $A$ and $B$ the temperature amplitude $\hat{T} = \sqrt{A^2 + B^2}$ and phase $\psi = \arctan(B/A)$ can be calculated. An 180°-ambiguity of the wave vector direction is solved by assuming all waves to be propagating upwards, i.e. $m$ is defined to be negative.

We choose to base our ray-tracing investigations on gravity wave fits at an altitude of 25 km as it is close to the base of the stratospheric jet. In addition, for this altitude the separation of gravity wave fluctuations and background is sound with relatively low cutoff zonal wavenumbers (Strube et al., 2020) and the wind gradient and, thus, the change of vertical wavelength with altitude is less pronounced than in the UTLS. Choosing higher altitudes would mean longer backward-trajectories and hence higher uncertainties. However, different altitudes mean a different state of lateral propagation, on the one hand, and critical-level filtering on the other hand. We will discuss this in section 4.5.

Sensitivity testing in previous studies (Lehmann et al., 2012; Preusse et al., 2012) indicates best performance of the S3D method if the cube size is in the middle of the range of expected wavelengths. Furthermore, the longest expected horizontal

wavelengths should not exceed three times the horizontal cube size. Therefore, we choose the extent of the fitting cube to be 35 grid points (7 °) in the zonal and 21 grid points (4.2 °) in the meridional direction. This corresponds to about 450 km×450 km horizontal extent at 55°S, and is guided by the range of 300 to 1000 km horizontal wavelengths identified in the temperature residuals above (cf. Fig. 2). From the temperature residuals, gravity wave vertical wavelengths are expected to range between 265    5 and 20 km. Therefore, we choose a vertical cube size of 10.5 km (21 grid points) for the fit.

Figure 3 shows horizontal maps of wave parameters for 25 km altitude obtained for the strongest sinusoidal wave component resulting from the S3D method in the investigated fitting volumes. Inspecting the temperature amplitudes (Figure 3a), there are three local maxima of wave amplitudes in the fits: one in the lee closely southeast of New Zealand's south island ((165°E,45°S) to (180°,55°S)), one further south-east of this first maximum ((175°W, 57°S) to (165°W, 62°S)) and one far to the east from 270    the first one ((160°W, 50°S) to (150°W, 55°S)). These are marked by ellipses in Fig. 3 and, for simplicity, the maxima will be referred to in the following as **I** (short for "Island"), **S** (short for "South") and **E**(short for "East") amplitude maximum, respectively. These maxima in temperature amplitude also show areas of maximum GWMF (see Figure 3b). GWMF depends both on the squared temperature amplitude and the ratio of horizontal to vertical wavelengths.

Horizontal wavelength values (Figure 3c) increase from the northeast to the southwest between approximately 200 km and 275    1000 km. The shortest horizontal wavelengths are found in the area of the **I** maximum and the longest around 60°S located at the **E** amplitude maximum. Vertical wavelengths (Figure 3d) increase from north to south ranging between about 5 km for region of the **I** amplitude maximum and 20 km at the **S** amplitude maximum. This is consistent with the fact that the wind maximum of the stratospheric polar vortex is around 60°S and refracts the waves there to longer vertical wavelengths. In general, the vertical wavelengths are 1 to 2 orders of magnitude smaller than the corresponding horizontal wavelengths.

The ground-based period, $\tau_{gb}$, shown in Figure 3e is the period of one full oscillation as observed from the ground. The quantity relates to the ground-based wave frequency $\omega_{gb}$ by $\tau_{gb} = \frac{2\pi}{\omega_{gb}}$. The intrinsic frequency $\hat{\omega}$ is calculated as a positive number per definition, but due to the Doppler shift, the ground-based frequency $\omega_{gb}$ and hence also the period $\tau_{gb}$ may take positive or negative sign. Positive values show movement of the gravity wave phase lines in direction of the wave vector, i.e. to the southwest and, analogously, negative values show movement to the northeast. Most of the wave structure has positive 285    wave periods (in particular in the **I** and **S** amplitude maxima) corresponding to periods of approximately 20 h. These relatively long, ground-based periods indicate almost stationary gravity waves consistent with mountain waves or other low ground-based phase speed sources. Negative ground-based periods are only found east of 160°W. The **E** amplitude maximum has negative $\tau_{gb}$ values that are smaller in magnitude than the positive values of the other two maxima and indicates waves with phase fronts moving eastward with respect to the ground.

The direction of the wave vector shown in Figure 3f represents also the intrinsic propagation direction and is perpendicular to the phase fronts as seen in the temperature residuals (Figure 2). Wave directions in the regions of the maxima (**I**, **S** and **E**) exhibit wave directions pointing west-southwest (≈210° counter-clockwise with 0° pointing to the east) to south-southwest (≈240°).

In summary, the three wave amplitude maxima (**I**, **S** and **E**) have each specific characteristics. The **I** amplitude maximum 295    is characterised by gravity waves with relatively short horizontal and vertical wavelengths, a wave direction pointing to west-

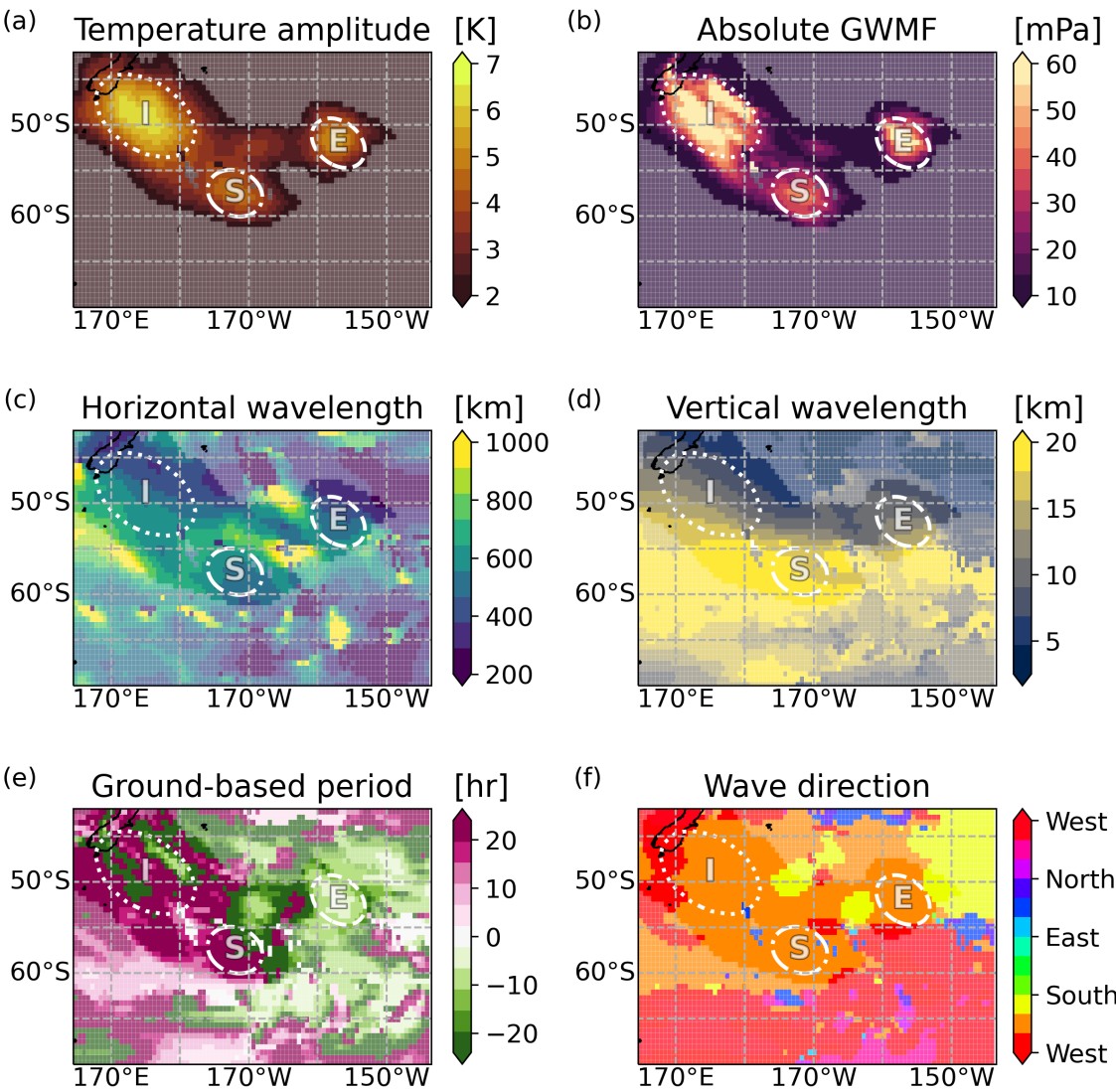

**Figure 3.** Horizontal maps of gravity wave parameters from sinusoidal fits in 35x21x21 grid point cubes applied on ECMWF-IFS temperature perturbations of 1 August 2014, 12:00 UTC. Cubes are centred around 25 km altitude. Hence, the panels show different gravity wave parameters for 25 km altitude: (a) temperature amplitudes, (b) GWMF, (c) horizontal and (d) vertical wavelengths, (e) ground-based period and (f) wave propagation direction. The letters in panel (a) refer to the locations of the different amplitude maxima - **I**="Island", **S**="South" and **E**="East" - as described in Sect.3.1. For amplitudes larger than 2 K the colours are as given by the respective colour bars (bright colours). For smaller amplitudes colours are weakened by a milky shading to indicate less significant wave events.

southwest, and an intermediate, positive ground-based period. The **S** amplitude maximum is characterised by waves with longer horizontal and especially long vertical wavelengths, wave direction pointing further to the south than for the other cases and very long wave periods. The **E** amplitude maximum is characterised by waves with also relatively short horizontal and vertical wavelengths, wave direction pointing to the southwest and a negative wave period.

For the backward ray-tracing described in the next section, a focus is set to the larger amplitude waves ($\hat{T} > 2\,\mathrm{K}$) for several reasons: As GWMF is proportional to the square of the gravity wave amplitude, these waves carry most of the momentum flux; the influence of imperfections in the background removal is less important for higher amplitude waves. In addition, gravity waves with amplitudes above a general background level can be expected to have a clear source attribution. Sources of gravity waves generate a wavenumber spectrum that depends on the physical dimensions of the source. Even if a source emits a

spectrum of waves, propagation to the altitudes we consider will spatially separate the spectral components. Accordingly, it has been shown by Lehmann et al. (2012) for the case of a typhoon that the wavenumber spectrum obtained with a spectral analysis method applied on a larger region around a source is well described by a set of small fit volumes covering the region, which each contain only a few components or even a single wavenumber. This facilitates backward ray-tracing to these sources which has been applied in a number of previous studies (Preusse et al., 2014; Krisch et al., 2017; Perrett et al., 2021). All

waves selected in this manner exhibit vertical wavelengths smaller than 2.5 times the vertical fitting-cube extent and horizontal wavelengths smaller than 3 times the horizontal fitting-cube extent.

### 3.2   Propagation paths

The Gravity wave Regional Or Global RAy Tracer (GROGRAT; Marks and Eckermann, 1995; Eckermann and Marks, 1996) is a ray-tracing model implementing gravity wave propagation based on the gravity wave dispersion relation and the ray-tracing

equations (Lighthill, 1978) under the Wentzel-Kramers-Brillouin (WKB) approximation. We apply GROGRAT to follow the propagation path of the stratospheric gravity wave packets, that are found (Sect. 2.3) and characterised (3.1) in the wave field, backwards in time to identify important propagation pathways and source regions for this case.

    GROGRAT is an established tool for studies on gravity wave propagation (e.g. Eckermann, 1997; Eckermann and Preusse, 1999; Preusse et al., 2002; Gerrard et al., 2004; Preusse et al., 2009a,b; Kalisch et al., 2014; Pramitha et al., 2015; Trinh et al.,

2016; Krisch et al., 2017; Yoo et al., 2020).

    The gravity wave ray-tracing equations describe the ray path and refraction along it by

$$\frac{d\boldsymbol{x}}{dt} = \frac{\partial \omega}{\partial \boldsymbol{k}} \tag{4}$$

$$\frac{d\boldsymbol{k}}{dt} = -\frac{\partial \omega}{\partial \boldsymbol{x}}. \tag{5}$$

In addition to the ray-tracing equations, an equation for the energy balance, defining the conservation of the wave action

density $A = \frac{E}{\hat{\omega}}$, with $E$ representing the total wave energy and $\hat{\omega}$ the ground-based frequency, completes the model. This assures the evolution of the amplitude within the wave packet to be represented. The software applies a saturation scheme

according to (Fritts and Rastogi, 1985) and also parametrisations for turbulent and radiative wave damping (Zhu, 1993) as the waves propagate upwards.

We use a version of GROGRAT originating from the status reported by Eckermann and Marks (1996) with an additional correction for spherical geometry in the refraction terms of the horizontal wavenumbers as suggested by Hasha et al. (2008) and implemented by Kalisch et al. (2014). The propagation allows for a slowly changing background represented by interpolation through snapshots of the background atmosphere.

GROGRAT applies time variation in the background atmosphere by adding a ray-tracing equation for the ground-based wave frequency $\omega$ (Eckermann and Marks, 1996)

$$\frac{d\omega}{dt} = k\frac{\partial U}{\partial t} + l\frac{\partial V}{\partial t} + \frac{((N^2),(k^2+l^2)-(\frac{1}{H^4}),(\hat{\omega}^2-f^2))}{2\hat{\omega}(k^2+l^2+m^2+\frac{1}{H^4})}. \tag{6}$$

Many previous studies use generic background atmospheres like standard atmospheres or temperature and wind profiles from climatologies for ray-tracing studies. The ECMWF operational analyses data provides a full atmosphere for the exact situation of this study. This provides a realistic, high-resolution background atmosphere, and we can investigate the ECMWF-IFS temperatures in order to understand the mechanisms that lead to the excitation of the waves in the model (Preusse et al., 2014). In particular, we launch GROGRAT rays from the wave parameters identified in the gravity wave characterisation introduced in Sect. 3.1 (i.e., the gravity waves resolved in the ECMWF-IFS operational analysis). Thus, there is full consistency between the waves that are traced with the ray-tracer and the background through which the they are propagated.

In general, GROGRAT rays may be terminated while tracing backwards because of three different reasons: (1) the rays may reach the ground, (2) the rays may approach a critical level from above and, hence, stall vertically (i.e. the vertical group velocity falls below a threshold of $0.01\,\mathrm{ms^{-1}}$), or, (3) the wave amplitude may vanish because of saturation. For the latter two criteria, the wave could exist at the ray-termination altitude only with insignificant amplitude which would not be compatible with large observed amplitudes at launch. The real source therefore must be located along the ray-path, but for (2) and (3) above the ray-termination altitude (Preusse et al., 2014).

### 3.3 Categories of backward-traced rays

We analyse the temperature perturbations in an analysis grid of overlapping fitting cubes every $0.6°$ in zonal and meridional direction. This yields more than 2 000 wave characterisations in the considered region. The analysis is limited to rays with a launch amplitude of more than 2 K. This leaves 1 280 rays covering the strongest parts of the structure, including the regions of the three temperature amplitude maxima discussed in Sect. 3.1 (cf. the regions marked by **I**, **S** and **E** in Fig. 3). The ray launch and termination points are shown in Figure 4 also featuring the areas of temperature amplitude maxima **I**, **S** and **E** in the launch point plots. All rays were launched from an altitude of 25 km altitude. The altitude of the termination points differs for each ray and is shown by the colour of the location dot in the termination point plots.

The altitude and location of ray-termination give indication of source processes which potentially excited these waves (Yoo et al., 2020). For a better overview, the rays are therefore categorised according to indications of different likely source processes described in this section and are shown accordingly in the three rows of Figure 4.

We have chosen two criteria to select rays: First, we screened along the ray path for a pass-by close to mountain ridges. If there was an elevation, that is higher than $500\,\mathrm{m}$, detected in a box of $2\,^\circ$ distance in longitude-latitude extent centred around and less than $1\,\mathrm{km}$ below the ray location, the ray was stopped there and assigned to the first category. In the following, this category will be referred to as "mountain" rays. "Mountain" rays do not necessarily represent waves that are classically generated by flow over a mountain ridge. Also stagnation flow in front of a mountain range and increased convection from corresponding cloud

formation, for instance, can trigger gravity waves (Galewsky, 2008; Houze, 2012). However, such processes are still closely related to the presence of mountain ridges. This is the reason why also rays that terminate in some distance from significant ridges are assigned to the "mountain" ray category. We will discuss the characteristics and how they relate to classical mountain waves in Sect. 4. Second, we collected all rays that are terminated above 5km altitude in the "high-terminated" category. When rays approach a critical level from above the vertical group velocity approaches zero, the saturated amplitude vanishes and the

ray is terminated accordingly. In this case, efficient energy transport from a lower level is not possible, hence, the source must be located above. We choose the altitude of 5km for the definition here to restrict the options for possible source processes for the corresponding waves. In the analysis area, an orographic wave cannot originate from above 5km altitude, since the orography in the New Zealand and Australia does not exceed 4km (The highest peak of New Zealand and Australia is the Aoraki/Mount Cook with 3724m). Furthermore, frontogenesis is usually diagnosed at the 850 hPa level, also below 5km.

Another possible option might be secondary wave generation, which, however, is not very likely in the lower stratosphere[1]. This leaves jet-related generation and deep convection as the most likely candidates for sources to the "high-terminated" rays.

    As a third category, we then collected the remaining waves, which will be referred to as "low-reaching rays". The number of likely source processes for this category is broader, since the generation can, in principle, take place anywhere along the ray.

    The three ray categories do not necessarily coincide with the three regions of temperature amplitude maxima (**I**, **S** and **E**)

discussed in Sect. 3.1. However, the local characteristics of the wave parameters, such as wave frequencies, are associated with different source processes in Sect. 4.2.

    Figure 4 shows in the left column the ray launch points in the stratosphere (at $25\,\mathrm{km}$) and in the right column the ray termination points respectively for the "mountain" (upper row of Fig. 4), "high-terminated" ray category (middle row of Fig. 4) and "low-reaching" (lower row of Fig. 4). The ray launch point marks the location of the relevant cube centres from the

S3D analysis discussed in Sect. 3.1. The total number of rays in each category is shown in the upper right corner of the ray-launch-point panel (left column). The grey dots mark the launch locations and their area is chosen proportional to the launch GWMF inferred for the individual ray. This highlights the locations that are most relevant for the momentum budget. The ray-termination point refers to the location of the lowermost altitude an individual ray reached in its propagation. In the panels showing the ray-termination points (right column), the percentage of the total GWMF at $25\,\mathrm{km}$ attributed to this category

---

[1]In the literature, there are no studies of secondary gravity wave generation for altitudes below $40\,\mathrm{km}$ and to be relevant in our case the generation would have to take place well below $25\,\mathrm{km}$

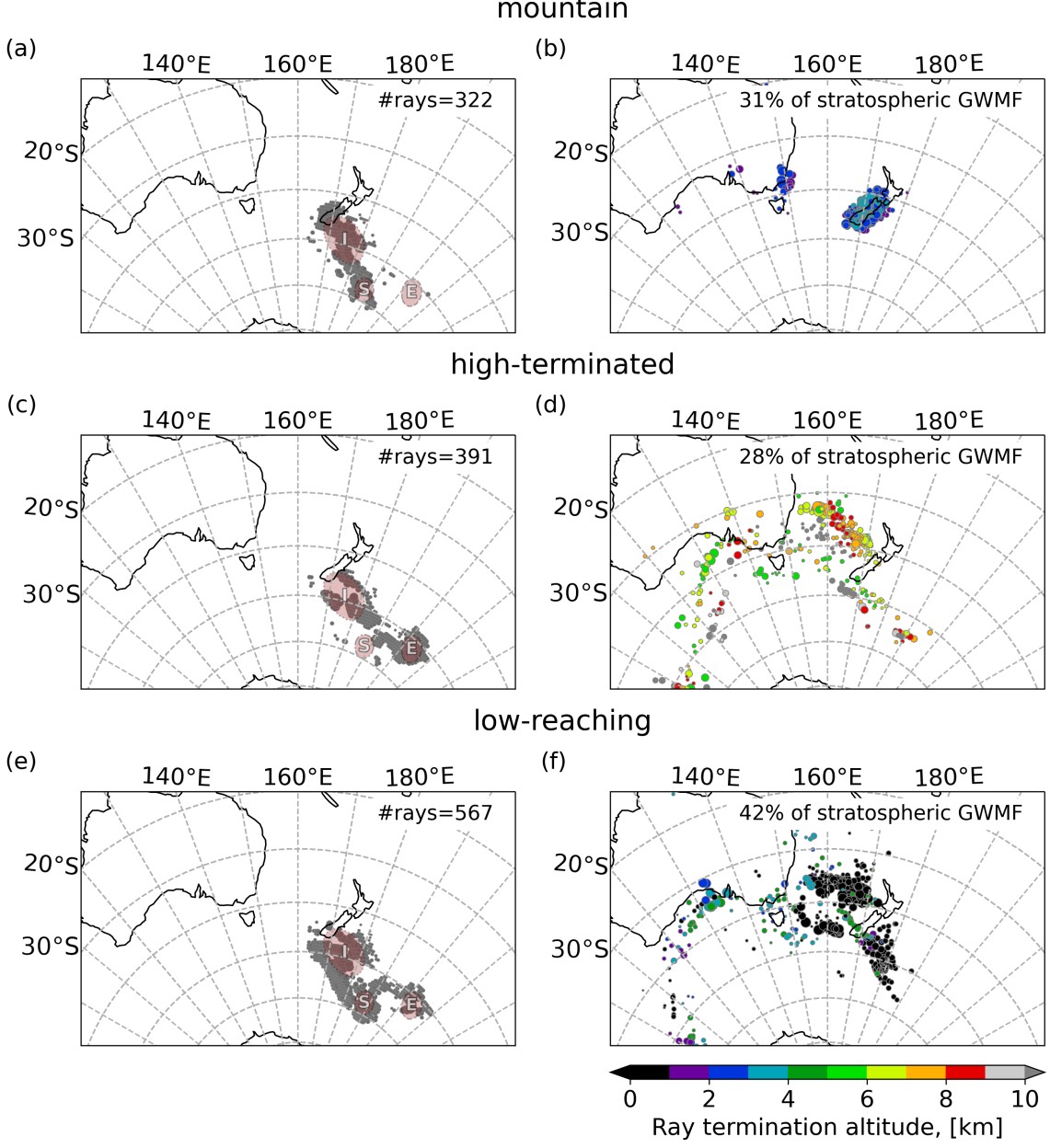

**Figure 4.** Ray-tracing results for (upper row) "mountain", (middle row) "high-terminated" and (lower row) "low-reaching" ray categories. Right column: Location of the ray-termination for different source categories. The colour of the markers shows the lowest ray altitude. Upper left corner: Percentage of stratospheric GWMF. Left column: Location of the ray launch points at 25 km altitude. Upper right corner: Total number of rays in that category

is shown. The coloured dots mark the location of the ray-termination points and the colour code shows the altitude of ray termination. Furthermore, the size of the dots is scaled with the inferred GWMF at the launch location of the ray analogous to the dots of the corresponding launch points on the left. This gives a visual aid which ray excitation areas are most important for the momentum budget in the stratosphere.

Most of the rays cluster locally both at launch and termination altitudes. At termination altitudes, the "mountain" rays originate mainly from around the Southern Alps on the South Island of New Zealand. Some are accumulating in the strait between the North Island and South Island. Furthermore, none of the "mountain" rays where launched from the **E** area which indicates that this group has a very specific propagation path. A few of the rays also trace back to Tasmania and mainland Australia; however, the associated momentum flux is small. In total, "mountain" rays in this case correspond to about one third (31%) of the sum of stratospheric GWMF determined at the launch level of 25 km altitude. "Mountain" rays, however, make for only about a quarter in number, 322 of the total of 1280 rays, which shows that they carry above-average GWMF. At launch altitude the locations for "mountain" rays (Figure 4 a) cluster in three areas:

1. around the South Island of New Zealand,

   $\rightarrow$ This is over the Southern Alps extending into the upwind-side of the mountain range.

2. in a streak extending from the tip of the island eastward for about 10° over the ocean, and

   $\rightarrow$ This coincides with large parts of the **I** amplitude maximum discussed in Sect. 3.1.

3. around 175°W between 55°S and 60°S.

   $\rightarrow$ This region coincides partly with the **S** amplitude maximum and indicates that "mountain" rays contribute to this structure.

The termination points of "high-terminated" rays cluster along an almost straight line above the Tasman Sea aligned from the northwest to southeast, approximately between (160°E, 30°S) to (170°E, 38°S). Most rays terminate there between 6 km and 9 km altitude. A second cluster is aligned south of New Zealand along a line from (160°E, 48°S) to (175°E, 50°S). Furthermore, there are a few small clusters at (145°E, 46°S), at (175°E, 48°S) and at (172°W, 52°S). The group at 100°E, south of 50°S corresponds to rays that leave the considered horizontal domain. "High-terminated" rays are associated with less than a third of the stratospheric momentum flux (28%). The launch areas cluster mainly on the **I** amplitude maximum and the **E** amplitude maximum.

Many of the "low-reaching" rays are launched in the western part of the gravity wave field directly south of New Zealand, i.e. as part of the **I** amplitude maximum, as well as from **S** amplitude maximum. The majority of these rays terminates below the core of the tropospheric jet upwind of New Zealand. For reference, Figure 1 shows the location of the tropospheric jet 24 h before the ray launch time (panel (a)) and at launch time (panel (b)). In particular for the **S** amplitude maximum, the waves originate at a source very close-by and propagate close-to vertically. "Low-reaching" rays are the largest class, both in GWMF (42%) and total number of rays (567).

In general, it is evident that for almost all rays the termination areas (indicating the source locations) are by far further north than the main GWMF patterns observed in the stratosphere. This will be discussed in more detail in Sect. 4.1.

## 4 Discussion

### 4.1 Lateral propagation dependent on wind and wave directions

In order to quantify the lateral propagation of the gravity waves, Figure 5 shows the total distance and the zonal and meridional shift separately for the three ray-termination categories. The left column shows the total distance the waves propagate between the ray-termination point in the troposphere and the ray-launch point at 25 km altitude. As already indicated by Ehard et al. (2017), some of the mountain waves propagate almost vertically in the troposphere and most of the stratosphere. Consequently, the "mountain" ray category (Figure 5a) is the only one which shows a major contribution also in the bin of shortest propagating distance which collects distances from zero to 500 km. In general, "mountain" rays travel short distances and hardly any are propagating further than 2 000 km away from the launch region. This is consistent with the modelling study of Jiang et al. (2019), who find for another gravity wave event of the DEEPWAVE campaign trailing waves from New Zealand to propagate onto the open ocean in the stratosphere. These trailing waves travel only about 10° southeastward of the island, and the main part of the wave field still remains over the island (Jiang et al., 2019).

"Low-reaching" (Fig. 5 c) and "high-terminated" (Fig. 5 e) rays are separated into two groups: one with propagation distances less than 2 000 km and the second with distances around 6 000 km. The relative importance between these two groups is shifted. For the "high-terminated" rays, longer distances occur more often while, for the "low-reaching" rays, the shorter distances seem to be more frequent. The result is consistent in a way, that "high-terminated" rays are closely connected and shaped by the critical level they approach, which usually means that they have lower intrinsic phase speeds and are more prone to lateral propagation. Furthermore, a large number of the "low-reaching" rays travel about 500 km to 1000 km distances, which is consistent with the overlap of launch and termination areas for the **S** amplitude maximum.

The right column of Figure 5 shows the zonal and meridional propagation distance. These propagation distances are measured in degree and, hence, the identical numbers in meridional distance compared to zonal distance indicate almost double the total distance in kilometres. For all three ray categories substantial southward propagation is evident. Notably, the largest meridional propagation is observed for the "low-reaching" rays despite the fact that "high-terminated" rays are propagating the larger total distance. Apparently, the "high-terminated" rays are drifting downstream with the wind while the "low-reaching" rays have a tendency for southward propagation.

We further investigated the latitude-altitude propagation paths, shown in Figure 6, in order to identify the altitude at which lateral propagation takes place. For all three termination categories, the gravity wave origin is mainly associated with the tropospheric wind maxima between 30°S and 50°S.

Gravity waves in the "mountain" ray category form two branches, a compact one propagating almost vertically and a second, spreading branch which shows southward propagation below 20 km altitude. The main paths are flatter below 15 km altitude

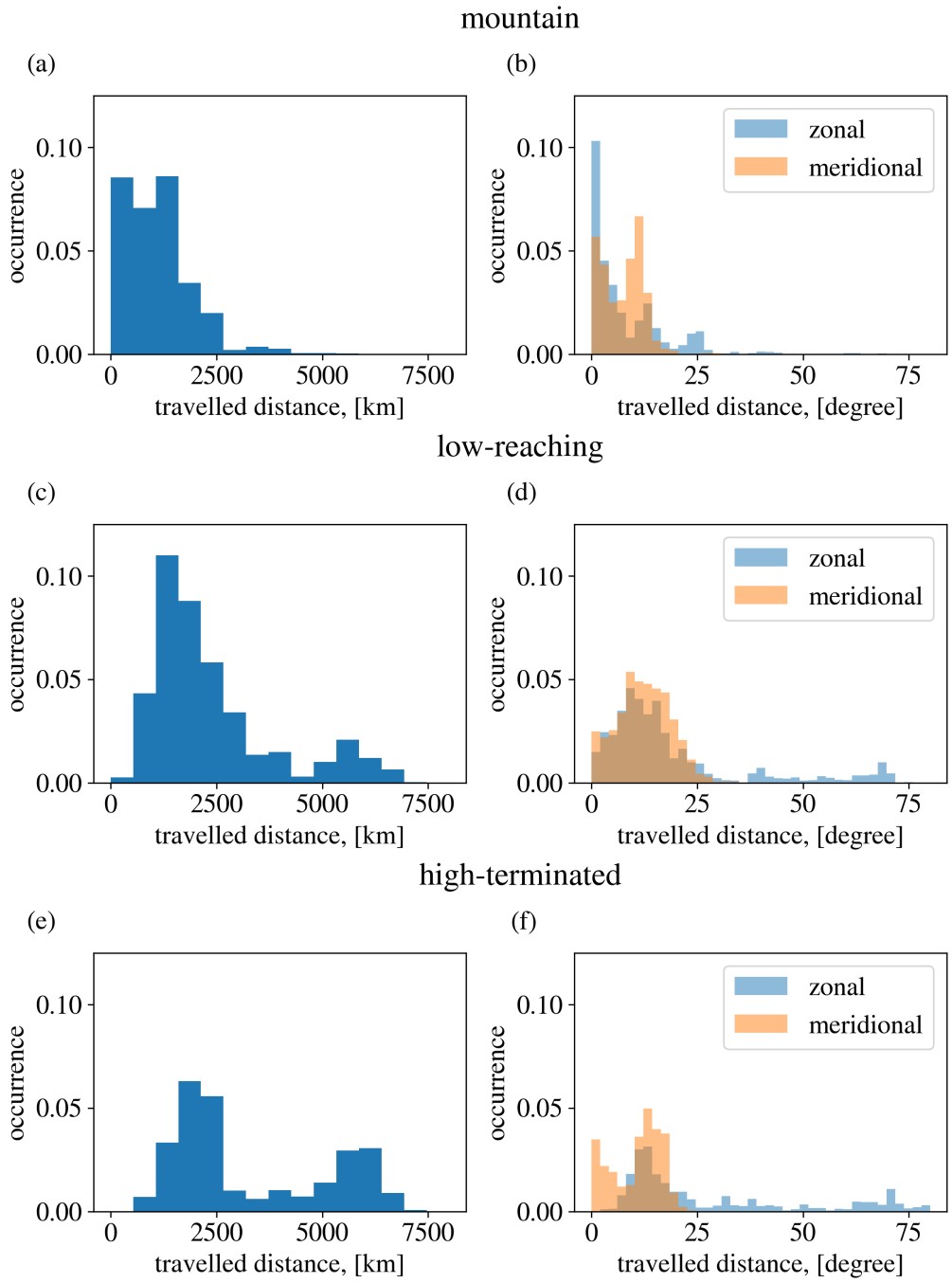

**Figure 5.** Distribution of distance between launch point (25km altitude) and termination point below. Distances are weighted by relative GWMF at ray launch, hence rays associated with more momentum flux in the stratosphere are emphasised. The left column shows geometric distances along a great circle; the right column shows distances in zonal and meridional direction.

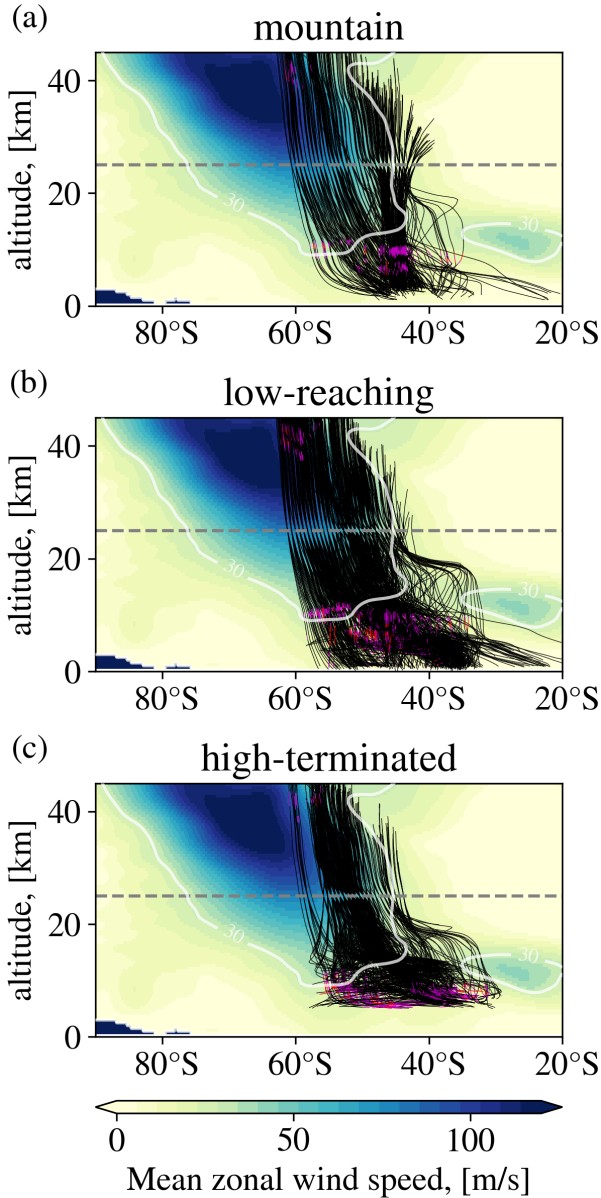

**Figure 6.** Latitude-altitude location of backward ray traces. The magenta and red colored sections of the rays show where the vertical WKB criterion is violated. The colour code in the background shows the mean zonal background wind on 1 August 2014, 12:00 UTC. The white contour line on top of the rays marks the 30 $\frac{m}{s}$ mean zonal wind to highlight the shape of the jet covered by the overlaid ray paths. The grey dashed line marks 25 km altitude, the launch altitude for the ray tracing experiment.

and steepen above. A similar steepening behaviour was also shown in Krisch et al. (2017) for a mountain wave packet from
Iceland in the northern mid to high latitudes.

"Low-reaching" rays show the furthest meridional propagation (cf. also Figure 5). A common pathway runs from about 35°S close to the ground to 55°S (or further south) at 15 km altitude. Above, also these rays steepen and proceed to propagate close-to vertically once they enter the stratospheric jet.

Lastly, "high-terminated" rays tend to experience a very flat propagation at lower altitudes almost propagating only horizontally for a time. Then, these waves show very low vertical group velocities indicating a state very close to vertical stalling. This points to waves with small vertical wavelengths already in the lower stratosphere (around 15 km altitude), which is often associated with strong shears. This might also be an indication that the gravity wave source is actually located at the shear regions rather than the final ray termination point.

It should be noted that only those waves which enter the stratosphere are part of this study, because the ray tracing is performed from 25 km altitude backwards. In addition to these waves, there could be an abundance of gravity waves which propagate upward from mid-latitude sources, but reach a critical level, for instance in the tropospheric subtropical jet (20°S to 40°S) above 15 km altitude. These waves then would not contribute to the wave fields in the stratosphere. In the introduction, we highlighted an apparent contradiction between the modelling study of Holt et al. (2017), who see momentum flux maxima mainly at mid-latitudes for 15 km, and the superpressure balloon observations of Hertzog et al. (2008) and Jewtoukoff et al. (2015), who observe momentum flux maxima associated with the winter polar vortex at 18 km altitude. It seems unlikely that all this momentum is transported laterally over this very small altitude range of only 3 km. However, if waves that remain very close to their sources and have little relevance for the stratosphere were dominant at 15 km altitude but disappear above, that would explain this apparent contradiction in the location of the GWMF maxima. Testing this theory would, however, require a modelling study of upward propagation from relevant sources which is beyond the scope of this study.

## 4.2 Relation between ground-based phase speeds and gravity wave sources

The ground-based phase speed is closely related to the source process, but also highly relevant for the propagation path of a gravity wave. In theory, gravity waves induced by flow over orography, i.e. mountain waves, are expected to have zero ground-based frequency in a constant incident flow on the mountain ridge. Changing wind velocities as well as possible interactions with clouds or turbulence (Worthington, 1999) are expected to induce slow non-zero ground-based phase speeds.

Figure 7 shows the GWMF-weighted distributions of the ground-based phase speeds among the three ray categories at launch and termination altitudes. Indeed, the "mountain" ray category shows the most compact of all three distributions centred around zero and mostly inside $\pm 10\,\mathrm{m\,s^{-1}}$.

Waves with low ground-based phase speeds have intrinsic phase-speeds closely related to the wind velocity and the relative orientation between wind and wave vector. For instance, if a wave of zero ground-based phase speed is directed strictly opposite to the background flow, intrinsic phase speed and wind compensate each other. Thus, the balance keeps the wave above the source. At an angle, the component of the phase velocity compensating the wind is smaller: the wave drifts with the wind, and simultaneously, there is a component perpendicular to the wind (in this case southward), which lets the wave propagate meridionally. In the case of mountain waves, the ground-based propagation takes place along the horizontal phase fronts of the waves (cf. Sato et al. (2012) and Appendix A).

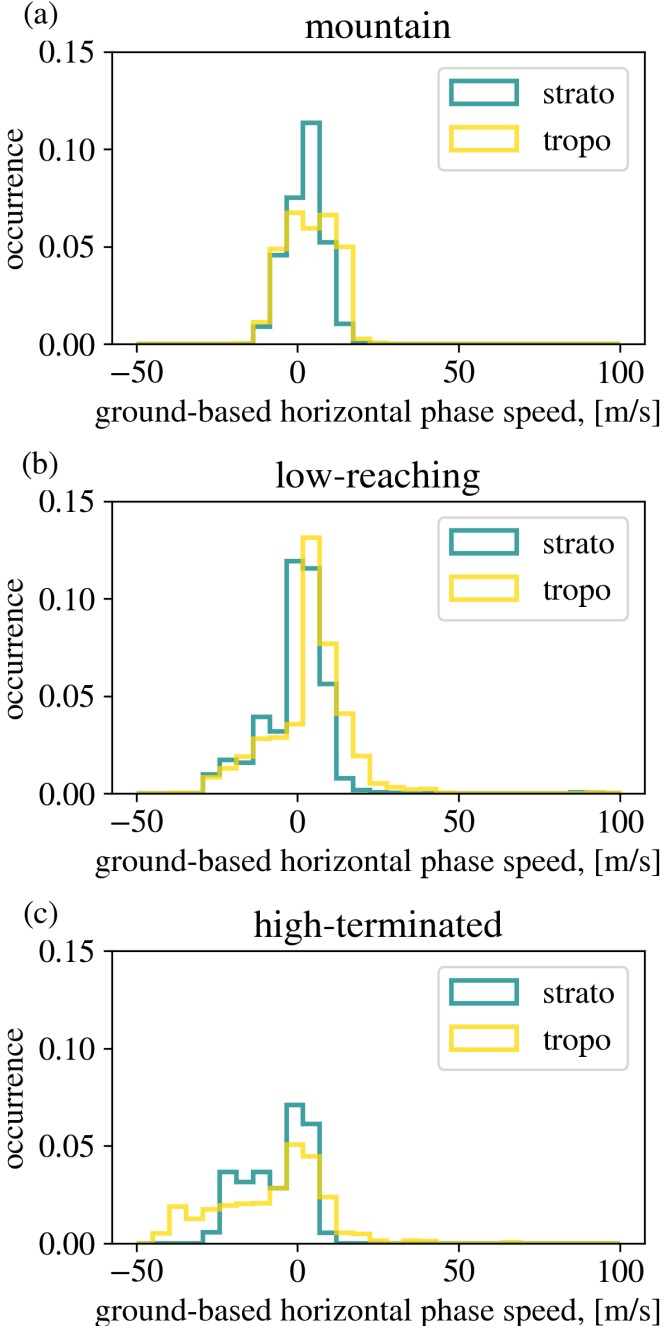

**Figure 7.** Phase speed of gravity waves in the direction of the wave vector and relative to the ground at (blue) observation altitude (25 km) and at (yellow) ray-termination for the three ray-termination classes, respectively.

The ground-based phase speed of the "high-terminated" rays is mostly negative with respect to the wave vector, i.e. drift and active propagation are in the same direction. This makes the "high-terminated" rays propagate very far laterally (cf. Sect. 4.1). The high, negative values occur especially at the ray-termination points at lower altitudes which also indicates that substantial lateral propagation already takes place at lower altitudes.

    The distributions of ground-based phase speeds for the "low-reaching" rays shows an intermediate between "mountain" and

"high-terminated" rays. The high number of rays with low ground-based phase speeds correspond well with the close locations of launch and termination points around the **I** and **S** amplitude maxima.

## 4.3   Angle of the waves relative to the ground and to the wind

The orientation of the wave vector indicates into which direction the wave propagates, while the relative orientation of the wave to the wind direction determines how far the wave propagates laterally. This is true especially for waves with low ground-

based phase speeds because the relative orientation determines the intrinsic phase speed and hence the vertical group velocity. Then, the drift with the wind is particularly effective for waves with a low vertical group velocity that stay at the same altitude level for a longer period of time. In Figure 8 this is investigated further by showing the propagation distance for the rays from termination to launch point on the vertical axis and the relative propagation direction, i.e. the angle between wave direction and wind direction, on the horizontal axis. We choose the altitude range between 6 and $18\,\mathrm{km}$ in order to investigate the altitudes

where the rays seem to undergo the majority of the meridional shift (see the gradient in the ray paths in Fig. 6).

    In general, almost all directions between $90°$ and $270°$ seem to exist. However, the closer the waves are oriented opposite to the wind ($180°$), the shorter are the travelling distances, at least for distances below $3000\,\mathrm{km}$ distance. As expected, for "mountain" rays there is an almost linear dependency between propagation distance and angle: rays with $180°$ relative propagation distance (i.e. wind and wave directed strictly opposite) are remaining over the source while the furthest propagation

for "mountain" rays is reached for approximately $225°$ (or $45°$ from opposite to the wind). Waves at steeper angles, i.e. greater than $225°$ are rather drifting downstream than propagating southward and are, in addition, easily dissipated.

    For the "low-reaching" rays, which possess non-zero ground-based phase speed, the similar relation is visible, but less pronounced. The "high-terminated" rays that are propagating very far ($\approx 6000\,\mathrm{km}$) are propagating at a variety of different angles. Angles between $90°$ and $180°$ are of little importance for waves propagating smaller distance and almost absent for

"mountain" rays.

    So far we have investigated the dependency of the propagation distance on the relative angle between background wind and gravity waves. Given that intrinsic group velocity is along the phase lines one may expect a similar dependency on the horizontal wavelength, i.e. that gravity waves with longer horizontal wavelengths propagate further. In our case, there is only a weak correlation, mainly caused by the mountain waves staying close to the island, which have also relatively short horizontal

wavelengths. These are also the waves which are, over a larger altitude range, directed opposite to the winds. Jiang et al. (2019) also found shorter horizontal wavelengths staying closer to New Zealand but for a different case. This hints to the fact that the excitation likely favours those short-scale waves that are directed approximately opposite to the winds. The finding could, however, also be linked to the particular topography of New Zealand where the main ridge is oriented South-West to North-East

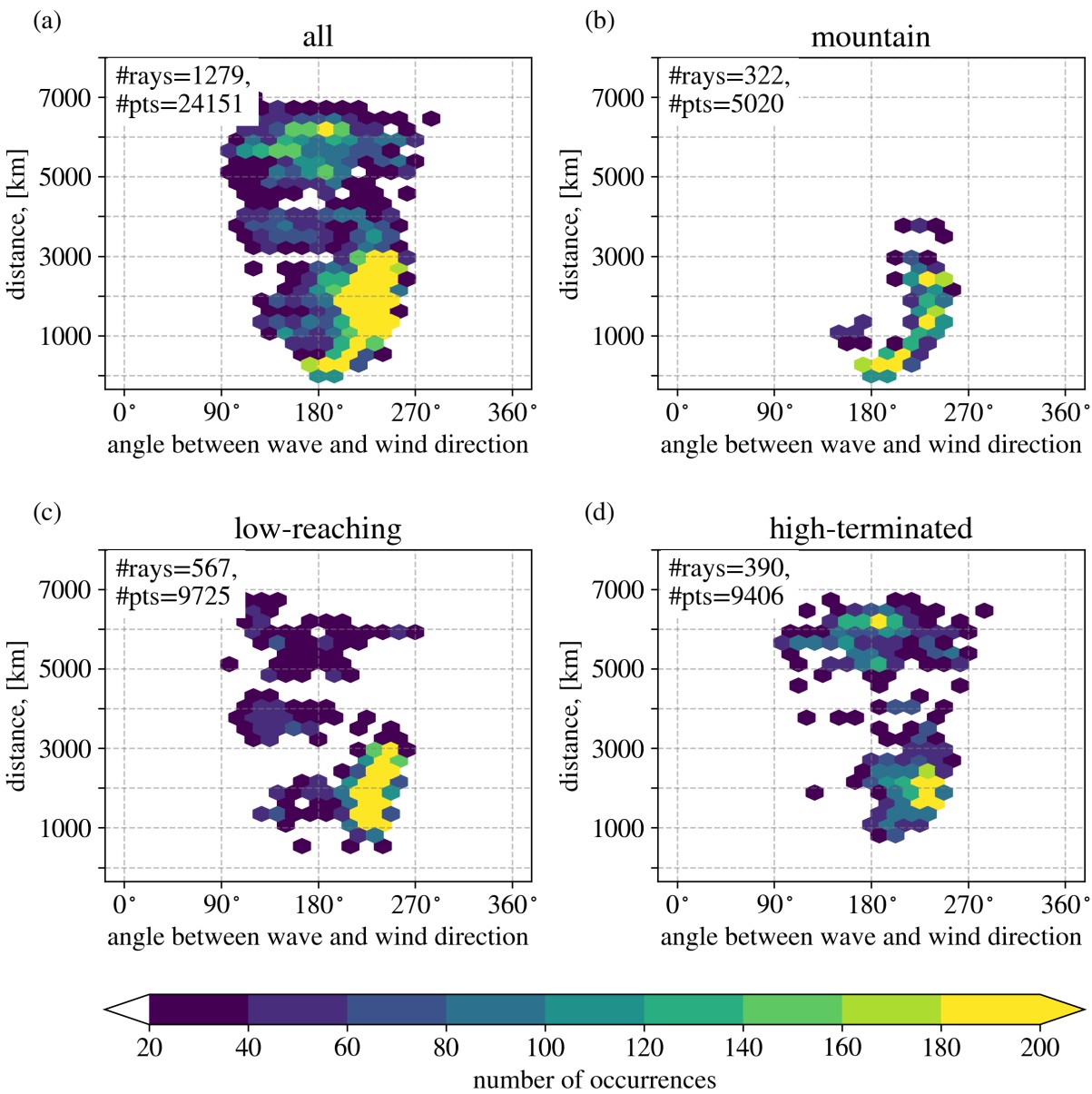

**Figure 8.** Relation of relative orientation of the wave to the wind direction and the distance of the corresponding ray from launch to termination point. Instances are picked at all ray locations between 6 and 18 km altitude to focus on the altitudes where most of the lateral propagation takes place. In the upper left corner of each panel the total number of rays (#rays) and the total numbers of instances (#pts) is stated. The colour code shows the number of instances in each bin. 180° indicates opposite orientation, the difference angle is counter-clockwise with respect to the wind direction.

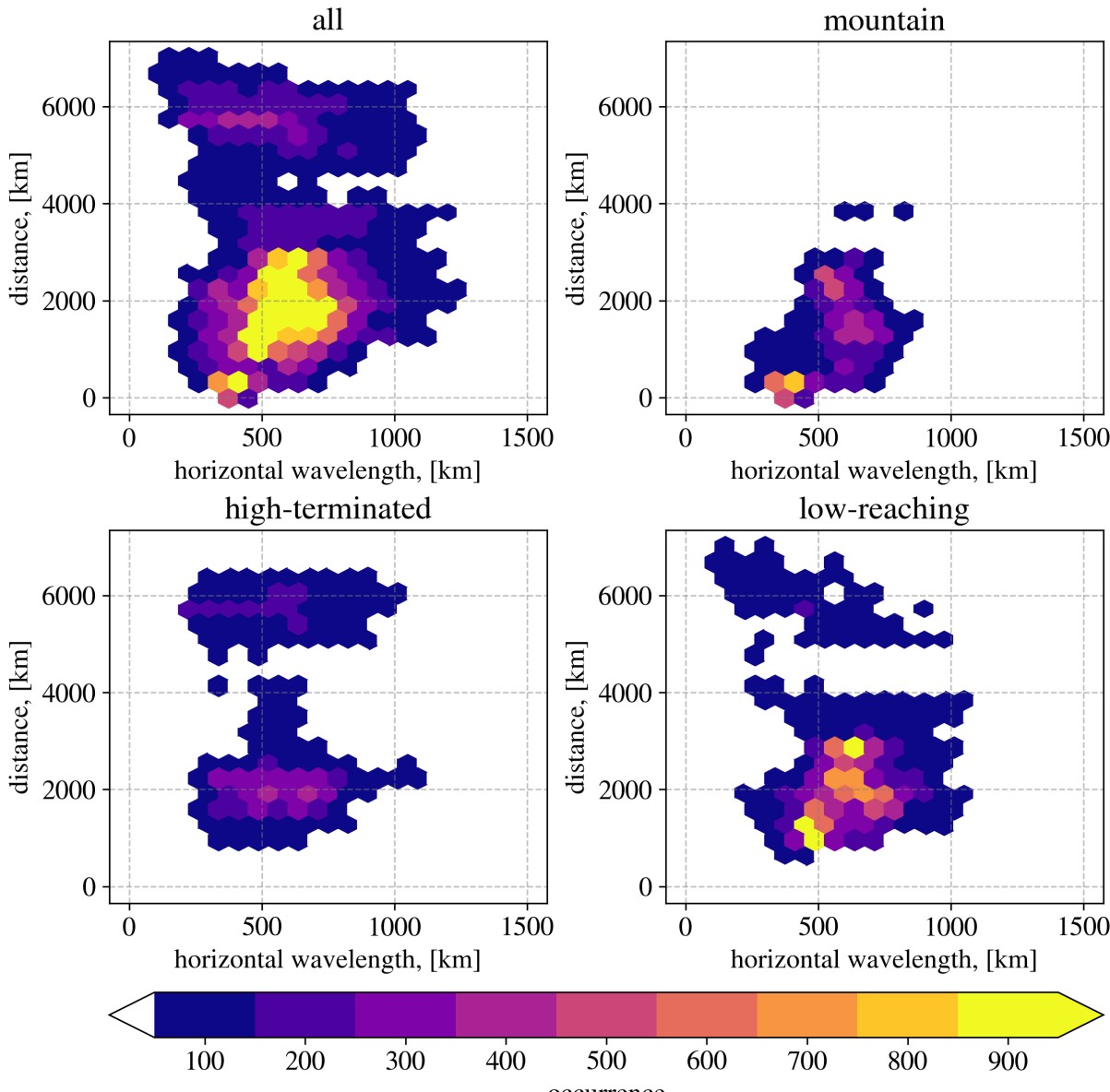

**Figure 9.** Same as Figure 8, but for horizontal wavelength on the x-axis.

and may be different for other mountain ridges, for instance Patagonia, where the main mountain ridge favours wave directions
facilitating lateral propagation into the stratospheric jet.

The results shown so far in this section suggest that the relative wave direction with respect to the wind is the governing factor for the lateral propagation distance of the low phase-speed gravity waves considered here. The question then is: What determines this angle? In previous studies, horizontal refraction was presented as a major factor for lateral propagation and

focusing of gravity waves into the stratospheric jet stream (Sato et al., 2009; Preusse et al., 2009a). On the other hand, the sources may generate a favourable direction of the gravity waves from excitation. What, hence is more important, the direction at excitation or the change of direction due to refraction?

In this context, Figure 10 shows the propagation direction relative to the ground in the left column, to the wind direction in the middle column and the relative angle between waves and wind as direction in the angular axis (measured counter-clockwise) in the right column. For the left and middle column, $0°$ points east, $90°$ points north and so forth. The radial component represents altitude from 0 to 25 km. The figure furthermore separates the results for the three ray categories "mountain", "low-reaching", and "high-terminated" rays in the three rows, respectively.

For the "mountain" rays (shown in the upper row, panels (a) to (c)), the majority of the waves point to approximately $225°$ (Figure 10a) for all altitudes and thus differs from the orientation of the main ridge, which would cause waves around $135°$. This orientation, hence, corresponds well with trailing waves as one important process included in the "mountain" ray category discussed repeatedly in the previous sections. The wind turns with altitude (panel b). This turning translates to the relative propagation directions (panel c), since the wave directions are mostly constant with altitude. However, the relative direction always remains smaller than $270°$, hence avoiding the directional critical level. There is a secondary, weaker branch of common directions around $150°$ in the wave directions (panel a) at low altitudes which turns with altitude to $180°$. These waves are excited with wave vector orthogonal to the mountain ridge. This branch has relative propagation directions closer to $180°$ (panel c) throughout all altitudes, because of the wind direction turning. Therefore, the waves remain mostly stationary to the ground. This relates to the mountain waves discussed by Ehard et al. (2017) where waves are kept stationary over the mountain up to the middle stratosphere and may then shift laterally into the jet. However, as already mentioned, this process is expected to be represented less dominantly in this study because of the selection of waves from the large wave field mainly south of New Zealand.

"Low-reaching" rays (middle row of Figure 10) mostly exhibit directions of approximately $225°$ to the ground. Unlike for the "mountain" rays, the "low-reaching" rays do not show a secondary group of mostly stationary waves with $180°$ direction, especially not for high altitudes.

"High-terminated" rays are the only group with a notable fraction of eastward propagating waves (panel g). The direction relative to the wind is generally opposite, that is, only the left-hand side of the circle is filled, but there almost all directions occur. In particular, for lower altitudes near 10 km, angles can be very close to $270°$, which distinguishes the high-terminated rays from the other two groups and in particular from the mountain waves.

## 4.4   Origin areas of non-orographic rays

The position close to the source altitudes of "low-reaching" and "high-terminated" rays with respect to the jet is shown in Fig. 11. The jet is plotted in horizontal wind maps for 18:00 UTC on 31 July 2014, a time step close to the time at which most rays, and in particular those closer to New Zealand, are terminated. "Low-reaching" rays are found often several 100 km upstream of New Zealand and closer to the jet core than the rays terminate higher up.

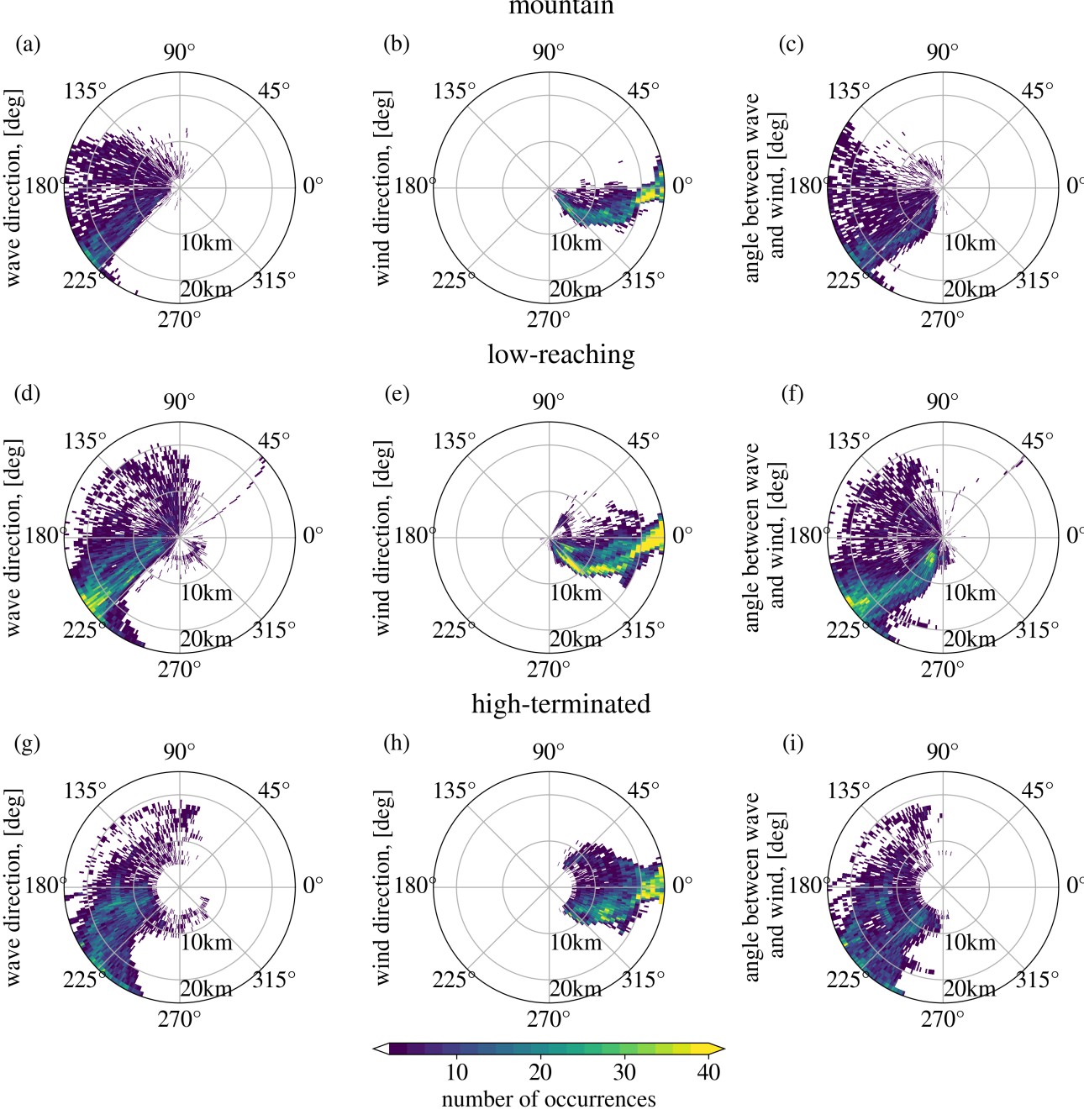

**Figure 10.** Polar maps of wind and wave directions along the GROGRAT rays binned into 0.5 km altitude (radial) and 1° boxes. The colour maps shows the number of ray instances in the corresponding box. The left column shows the propagation direction with respect to the ground, the middle column the wind direction and the right column the relative propagation direction to the wind.

"High-terminated" rays at $8\,\mathrm{km}$ altitude originate mainly from the northern border of the jet, a region of strong wind gradients. Their phase fronts are approximately parallel to the wind and seen also in tropospheric vertical winds (cf. Figure 12).

Figure 12 shows the relation of local wave properties of selected "high-terminated" rays together with the background atmosphere at $8\,\mathrm{km}$ altitude. The vertical wind maps show different types of perturbation all over the presented region, but especially between 150°S and 170°E. A streak of disturbances with small spatial scales extends southeast from the East coast of Australia (around [150°E, 35°S] to [165°E, 55°S]). Comparing the location of these disturbances with the horizontal wind speed maxima in Fig. 11, it is evident that they are located right in the centre of the subtropical jet. Vertical wind patterns that point to a jet-exit region can be found in the vertical wind map north of the jet stream (roughly between (155°E, 35°S) and (170°E, 40°S)). Most of the waves are found co-located with long horizontal wavelength wave structures of weak amplitudes. Again, the waves most relevant for the stratosphere are not necessarily the strongest waves but those which are able to enter the stratosphere and hence are contained in our study. A detailed discussion of the source processes is however beyond the scope of this study.

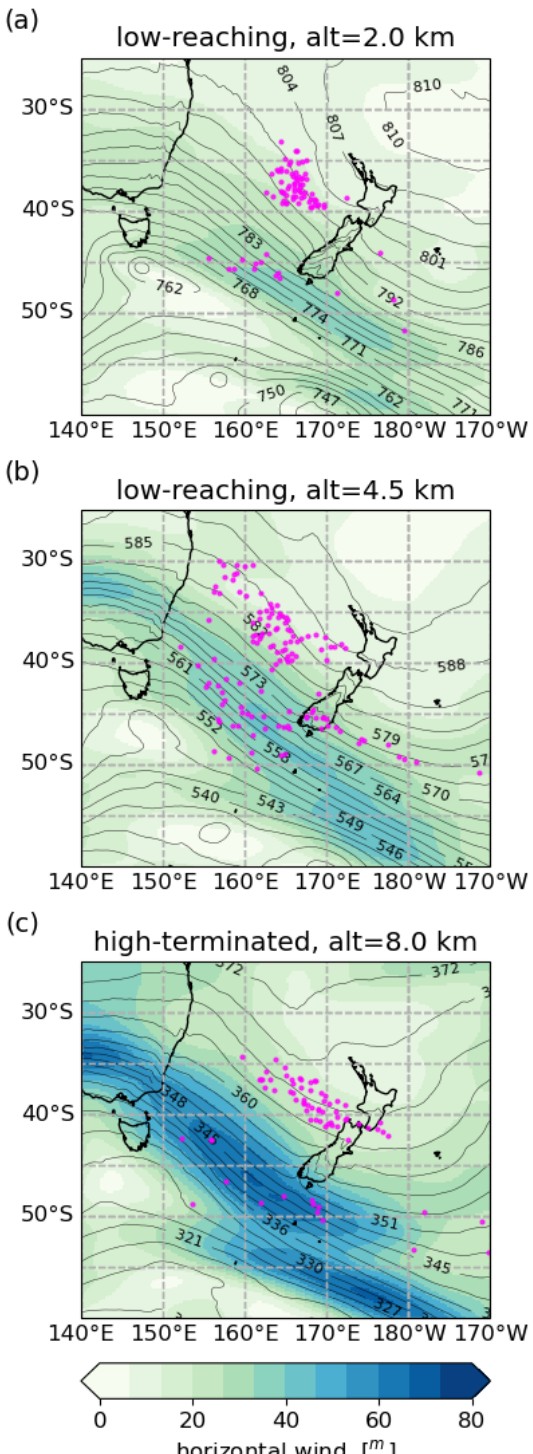

**Figure 11.** Current ray locations on horizontal background wind maps (i.e., large scale part from the scale separation) from ECMWF operational analyses of 31 July 2018, 18:00 UTC signified by the colour code. Contours show the pressure levels with values specified as numbers on the contour.

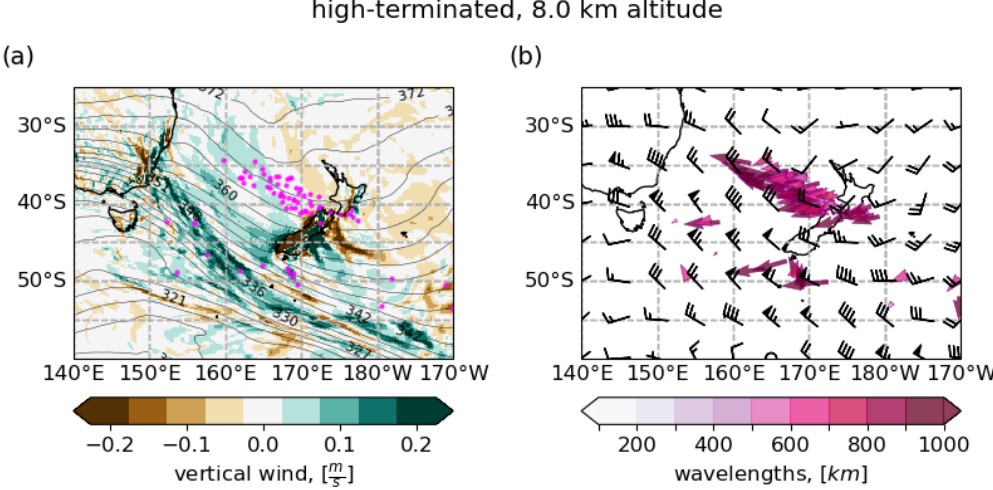

**Figure 12.** On the left: Position of ray-termination points overlaid on vertical wind velocities for 31 July 2014, 18:00 UTC, i.e. 18 h before the wave patterns where analysed at 25 km altitude. Only ray locations that coincide with altitude and time are shown. On the right: Arrows show the direction of the wave vector at the ray location. The colour code highlights the different horizontal wavelengths of the shown rays.

## 4.5 Dependence on the investigation altitude: effects of gravity wave filtering

Throughout this study, we have investigated the origin and properties of waves around 25 km altitude. For reasons of wave analysis and ray-tracing errors, we consider this an optimal altitude. Furthermore, most of the lateral propagation has occurred below this altitude and it is close to the base of the stratospheric vortex. However, the question remains how filtering affects the results. In order to investigate this, we have conducted ray-tracing experiments based on S3D analyses for 20 km and 30 km cube-centre altitude (Figures 13 and 14). The S3D fits that were used as launch conditions of the raytracing at 20 km and 30 km were generated with ECMWF field from 1 August 2014, 06:00 UTC, (6 hours before the fits at 25 km) and 1 August 2014, 18:00 UTC, (6 hours after the fits at 25 km), respectively, to assure that we capture approximately the same wave packets at the different altitude. Otherwise the S3D fitting was perform with the same settings as described in Section 3.1.

As expected from the discussions in the previous sections, clusters of gravity wave momentum flux are located further north relative to the positions at 25 km altitude for 20 km analysis altitude. The corresponding southward shift from 25 km to 30 km analysis-altitude is much smaller. This is consistent with our finding that the ray paths steepen in the stratosphere. In general, the patterns of the excitation locations are most compact for 20 km launch altitude, while for 30 km launch altitude a larger spread is found. This is to be expected, as the ray trajectories become longer and errors grow along the trajectories. Despite this fact we still recognize largely the same general source regions throughout all altitudes, but with major shifts regarding their relative importance. At 20 km altitude we still find a large contribution of waves propagating very obliquely. This is expressed in a high contribution from the high-terminated rays as well as far downstream propagating mountain waves from Tasmania. For 30 km altitude the most striking feature is the loss of importance of mountain waves (only 16% instead of 31% at 25 km). As can be seen from Figure 6 many of the mountain waves which stay above the island encounter a critical level between 25 km and 30 km altitude and mainly the trailing waves propagating further to the south survive. At 30 km altitude also sources from the south and further away contribute, so we see a high total number of rays. The wider distribution also seems to hint at a general tendency that gravity waves from strong sources become less important and give way to a more unspecific background of gravity waves from a large variety of source locations. In addition it suggests that waves from the south, at least in our situation, enter the vortex at higher altitudes.

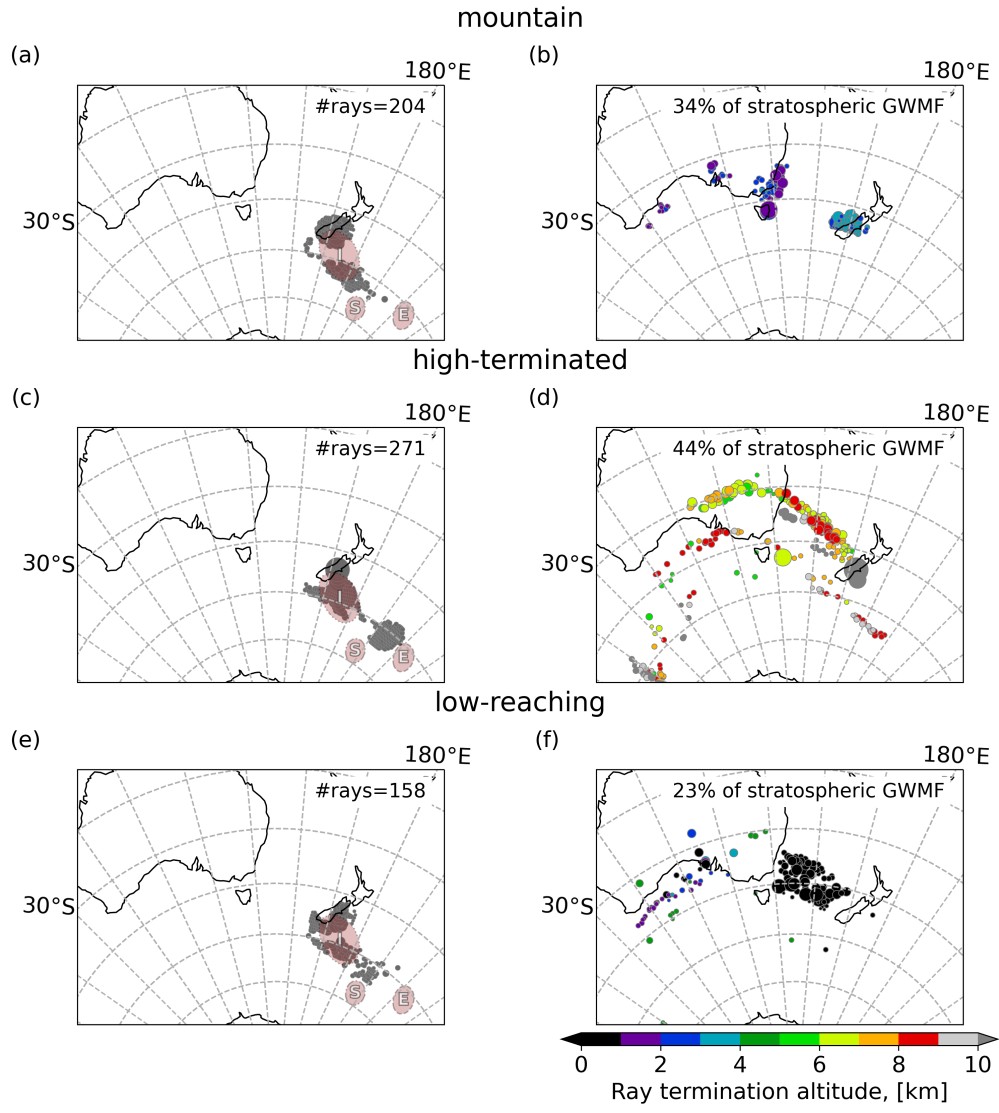

**Figure 13.** Same as Figure 4, but for an S3D-analysis altitude of 20 km

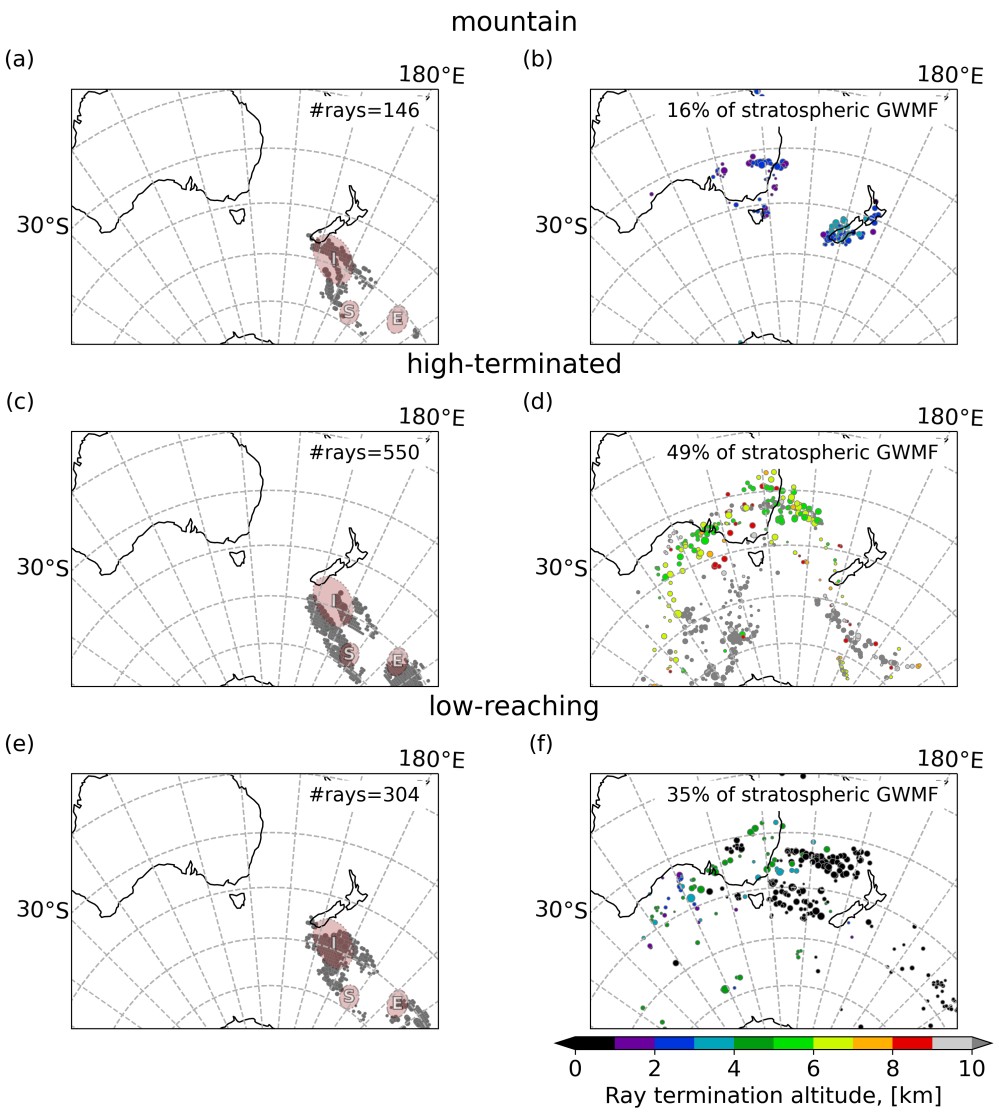

**Figure 14.** Same as Figure 4, but for an S3D-analysis altitude of 30 km and a minimum amplitude of 3K (instead of 2K at 25 km altitude)

## 5 Summary and conclusions

One of the hypotheses for the origin of GWMF in the southern polar jet at approximately 60 °S is pole-ward propagation of gravity waves from mid-latitude sources around 40°S (Wu and Eckermann, 2008), thus requiring gravity waves to propagate laterally by ∼20°.

Previous studies of GWMF propagation into the stratospheric jet predict a focusing of the gravity waves into the jet driven by horizontal refraction over a wide altitude range in the stratosphere (Sato et al., 2009; Preusse et al., 2009a). However, there is an apparent contradiction. On the one hand, high-resolution model studies like Holt et al. (2017) indicate that at 15km GWMF is still located close to midlatitude source regions. On the other hand, GWMF observations from superpressure balloons and satellites in the upper troposphere and lower stratosphere show enhanced GWMF in the southern polar vortex jet around 60 S already at 18km (Hertzog et al., 2008), or in the lower stratosphere (e.g., Ern et al., 2011; Geller et al., 2013).

We investigate a gravity wave field south of New Zealand that was first presented by Ehard et al. (2017) in AIRS observations supporting their lidar study for the DEEPWAVE campaign. ECMWF operational analysis data of the structure show the same overall wave patterns as seen in the AIRS observations. We, therefore, use the ECMWF operational analysis data for our investigation because the regular sampling without gaps and the good vertical resolution compared to AIRS observations facilitate wave analysis of the whole field with good accuracy and systematic ray tracing.

In Table 1, we collect the identified characteristics of different part for the wave field separated by the three defined regions of high temperature amplitudes and the categories the corresponding rays were sorted into. The overview shows that different wave packets build the complex wave structure under investigation. Furthermore, these characteristics are important for the different behaviours later observed for the rays that are traced backwards.

The overview in Table 1 highlights that the wave field southeast of New Zealand is quite complex, though it might look homogeneous on the first glance. The three areas with high gravity wave temperature amplitudes, that also show particularly high GWMF, exhibit characteristically different prominent wave properties. We traced each of these maximum amplitude areas back to at least two different likely source processes that are represented by the different ray categories. We find all three source categories – orographic generation, low and high non-orographic sources – in approximately equal parts for this wave field. Measured at 25 km altitude, waves from orographic sources reach horizontal distances of up to 2000 km from their source. Generation processes at the tropospheric jets, like spontaneous adjustment, govern distant parts of the wave field with horizontal distances of up to 6000 km between their source and stratospheric location.

Most of the lateral propagation takes place between 5 and 15 km altitude, and almost all below 20 km. The waves exhibit low vertical group velocities in the troposphere and lower stratosphere and, hence, allow for an efficient drift with the wind. The vertical group velocities grow rapidly as the waves reach the stratospheric jet and let the waves propagate predominantly in the vertical. Considering the altitude distribution of the propagation direction there is indication for considerable horizontal refraction only in the part of "mountain" rays which remain over New Zealand and closely downstream. However, most waves in this study have a southwestward propagation direction already at source altitudes. Correspondingly, waves that experience lateral propagation and are reaching the stratosphere already possess a southward component in low altitudes. In generalisation,

| | region **I** | region **S** | region **E** |
|---|---|---|---|
| "mountain" rays | | | |
| – hor. wavelengths | intermediate (400-600 km) | long (>500 km) | no rays |
| – gb. phase speeds | slow (0-10 m/s), mostly positive | very slow (0-5 m/s), positive | |
| – wave directions | west to southwest | west to southwest | |
| "low-reaching" rays | | | |
| – hor. wavelengths | intermediate (400-600 km) | long (>500 km) | intermediate (400-600 km) |
| – gb. phase speeds | two branches: 1) very slow (0-5 m/s), positive; 2) intermediate (5-15 m/s), negative | two branches: 1) very slow (0-5 m/s), positive; 2) slow (0-10 m/s), negative | fast (15-20 m/s), negative |
| – wave directions | west to southwest | west to southwest | southwest |
| "high-terminated" rays | | | |
| – hor. wavelengths | intermediate (400-600 km) | no rays | intermediate (400-600 km) |
| – gb. phase speeds | very slow (0-5 m/s), mostly positive | | fast (10-25 m/s), negative |
| – wave directions | southwest | | southwest |

**Table 1.** Characteristion of waves in temperature amplitude maximum regions **I**, **S** and **E** with respect to their ray category of "mountain", "low-reaching" and "high-terminated" rays. Abbreviations: "hor." = horizontal, "gb." = ground-based.

this would indicate that waves are generated in the mid latitudes, e.g. from the tropospheric jet, and propagate with a significant southward component into the stratospheric jet. This does not mean that all waves generated there have a strong southward component, but the other waves from this region are filtered at the top of the tropospheric jet. In the transfer altitudes both tropospheric and stratospheric jets are comparably strong. Thus, the horizontal gradient is too weak to induce substantial refraction. Our case study therefore suggests to shift the focus of investigation from waves that are horizontally refracted by strong wind gradients into the polar night jet to waves that feature southward orientation already at source altitude, which could be more important also in general.

Gravity waves which may have been excited but meet a critical level on top of the tropospheric jet, and thus do not enter the stratosphere, are not accounted for in this study. This may explain a strong dominance of southwestward wave orientation and it may also partly explain why Holt et al. (2017) see GWMF still close to the sources at $15\,\mathrm{km}$, but Geller et al. (2013) find the GWMF maximum in the polar jet as low as $18\,\mathrm{km}$ altitude.

*Data availability.* The ECMWF operational analysis fields are available directly through ECMWF (2015).

## Appendix A: Horizontal propagation of mountain waves along phase fronts

How does a mountain wave propagate laterally? Without loss of generality we can consider the propagation of a mountain wave (i.e. wave with zero ground-based phase speed) for wind from due west ($\boldsymbol{u_h} = (u, 0)$) and a ridge from northwest to southeast as sketched in Figure A1. The orientation of the ridge (black line) determines the direction of the wave vector $\boldsymbol{k_h} = (k, l)$ (dark blue arrow). The horizontal phase speed has the direction of the horizontal wave vector. In order to determine the value[2] we have to consider the projection of the wind (green arrows) to the direction of the wave vector (e.g. Preusse et al., 2002): $u_k$. Since we have chosen $v = 0$, only the x-component contributes to the scalar product $u_k = \boldsymbol{u_h} \bullet \boldsymbol{k_h} = u\frac{k}{k_h}$ with the magnitude $k_h := |\boldsymbol{k_h}| = \sqrt{k^2 + l^2}$. The intrinsic horizontal phase speed (red arrow) of the mountain wave is then in direction of the wave vector and opposite to the projected wind:

$$\hat{\boldsymbol{c}}_{\boldsymbol{\phi}} = (\hat{c}_{\phi x}, \hat{c}_{\phi y}) = -u_k \left( \frac{k}{k_h}, \frac{l}{k_h} \right) = -\frac{uk}{k_h^2} (k, l) \tag{A1}$$

In mid frequency, i.e. using the dispersion relation $\hat{\omega}^2 = \frac{k_h^2 N^2}{m^2}$, the intrinsic horizontal group velocity equals the intrinsic horizontal phase speed ($c_\phi = \frac{\hat{\omega}}{k_h}$). Finally, the ground based group velocity is the sum of the wind vector $\boldsymbol{u_h}$ and the intrinsic group velocity $\boldsymbol{c_g} = \boldsymbol{u_h} + \hat{\boldsymbol{c}}_{\boldsymbol{g}} = \boldsymbol{u_h} + \hat{\boldsymbol{c}}_{\boldsymbol{\phi}}$, here given in purple. As can be seen from Figure A1 this is the wind component perpendicular to the wave vector and hence parallel to the mountain wave phase front. We can test this by performing the scalar product of the ground based group velocity and the horizontal wavevector:

$$\boldsymbol{c_g} \bullet \boldsymbol{k_h} = (c_{gx}, c_{gy}) \bullet (k, l) \quad = \quad \left( u - \frac{uk}{k_h}\frac{k}{k_h}, -\frac{uk}{k_h}\frac{l}{k_h} \right) \bullet (k, l) \tag{A2}$$

$$= \quad uk \left( 1 - \frac{k^2 + l^2}{k_h^2} \right) = 0 \tag{A3}$$

---

[2]The phase speed definition in a 1D frame work is $c = \frac{\omega}{k}$. Note, however, that this can not be trivially extended to a vector system by dividing through the vector components (e.g. Fritts and Alexander, 2003). Instead we have to determine the axis of interest, determine the wavenumber on this axis, and multiply the so-gained value with the unit vector defining this axis. In order to determine the horizontal phase speed we, accordingly, use the horizontal components of the 3D wave vector as the horizontal wave vector, determine the horizontal wavenumber $k_h$ and further project all relevant quantities on the direction of the horizontal wavevector thus reducing the calculations to 1D.

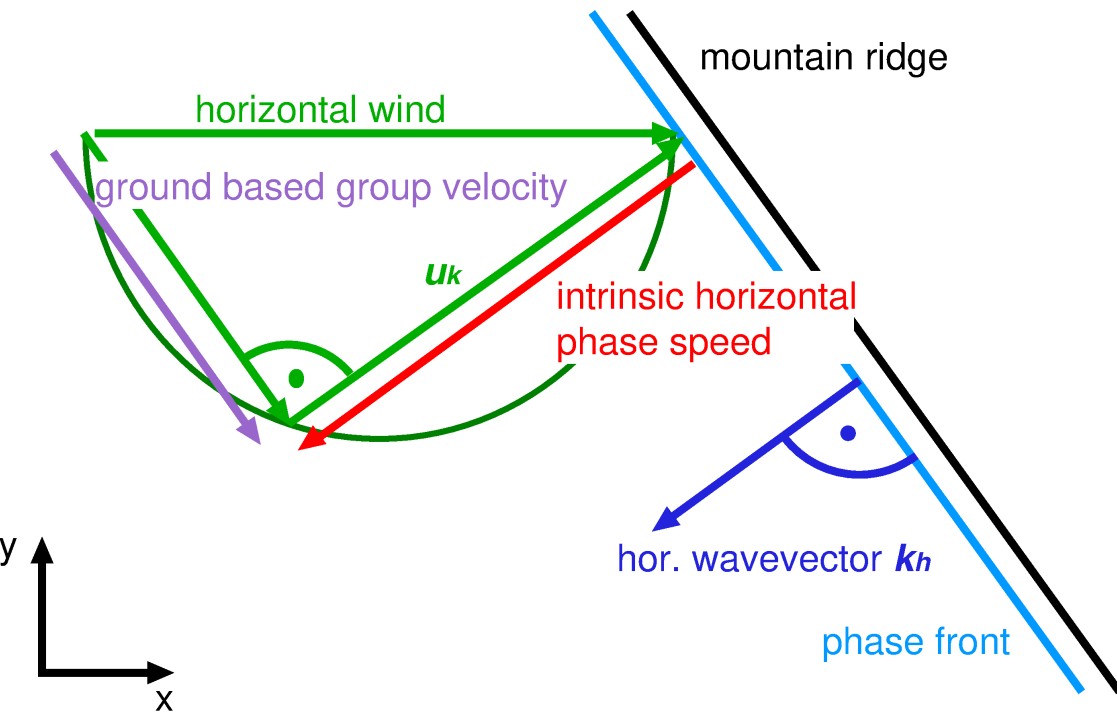

**Figure A1.** Schematic view of the horizontal propagation of a mountain wave. For details see text.

According to Thales's theorem, the green rectangular triangle of the wind is always inscribed in a semicircle with diameter $u_h$ and the southward component is largest, when the angle is 45°, i.e. $k = l$. Since the vertical group velocity $c_{gz}$ is lower for smaller horizontal phase speed, the wave will propagate most effectively to the south when k is slightly smaller than l but still close to 45°. Furthermore, the larger $\frac{l}{k}$, the more oblique is the propagation, but the smaller is also the horizontal phase speed, 665   the vertical wavelength, and, with this, saturation amplitude and saturation GWMF.

We have here sketched a ridge with northwest to southeast orientation as for instance for the southern Andes while the main ridge of New Zealand is oriented northeast to southwest. These waves can travel southward only after turning of the wavevector by horizontal refraction (cf. Ehard et al., 2017). However, also at New Zealand mountain waves with northwest to southeast orientation are generated, for instance trailing waves from the south tip of New Zealand, mostly due to the single peaks and 670   the edge of the ridge (Jiang et al., 2019). These also propagate along the horizontal phase lines as deduced here, as the only assumption made is the zero ground-based phase velocity. The relations will hold approximately for all waves where the wind is much faster than the ground-based phase speed, but not hold for waves with substantial ground-based phase speed such as waves from convection or shear instability.

**Appendix B: Consistency of gravity wave patterns inferred from different atmospheric quantities**

Linear theory implies that gravity wave disturbances observed in different atmospheric quantities like temperature, winds and vertical velocity are related by the polarisation relations (Fritts and Alexander, 2003). We concentrated our investigation for this study on one atmospheric quantity, temperature, since it is important especially for satellite observations. However, it has been established in previous research (Geller and Gong, 2010), that the choice of different quantities can emphasise different parts of the gravity wave spectrum in the analysis. The small-volume spectral decomposition method, S3D, used in this study

allows for a good balance of spatial localisation and spectral characterisation of the wave field. The S3D method is based on the scale separation between large-scale background and gravity wave fluctuations and the fit to data in a characteristic volume. With the choice of a specific volume size and the scale separation approach the method may additionally influence the wave spectrum included in the analysis. Therefore it is interesting to explore whether our choice of the quantity and method indeed have influenced our findings. We do that by investigating the consistency of the wave fields in the various quantities linked by

the polarisation relations.

Here, we follow two lines to check the consistency of our results: (1) we compare the perturbation fields generated by the background removal (scale separation with the approach described in Section 2.3) and perturbations reconstructed from the S3D fit results presented in Section 3.1 in different quantities – namely temperature, zonal and meridional winds as well as vertical velocity and (2) we show S3D results from vertical velocity analysis. In addition, we also compare the momentum flux

estimates from S3D with a different method. We also discuss the influence of using vertical wind instead of temperature on our inferred wave-origin locations.

**B1 Consistency of the perturbation fields**

The S3D method characterises a wave field by dividing the perturbation field into small sub-volumes and fitting a superposition of a monochromatic waves. This allows for a reconstruction of the wave field from the results using the fitting model: The data

grid is partitioned into regular volumes around the fitting cube center locations. Each volume is then filled with a superposition of sinusoidal waves calculated from the wave parameters for the corresponding S3D results. The corresponding fields for winds and vertical velocity can be calculated from the S3D results from the temperature perturbation fit by converting the amplitudes via the polarization relations and shifting the phases accordingly.

A comparison of input and reconstructed perturbation fields in the different atmospheric quantities is shown in Figure B1.

The left column shows the fluctuations of the respective quantity after background removal, while the middle and the right columns show the reconstruction from S3D temperature fields for two wave components superposed and only the first (strongest) wave component, respectively.

In value, variations in winds are about twice as large as in temperature, which is consistent with the polarisation relations. The large patterns of the wave field – like the wave fronts extending from the South Island of New Zealand southeast onto the

705 ocean – are well captured in location and magnitude by the reconstruction with one wave component already. The second wave component brings additional detail into the patterns; e.g. the structure in the east of the wave pattern (around 160° W, 50° S)

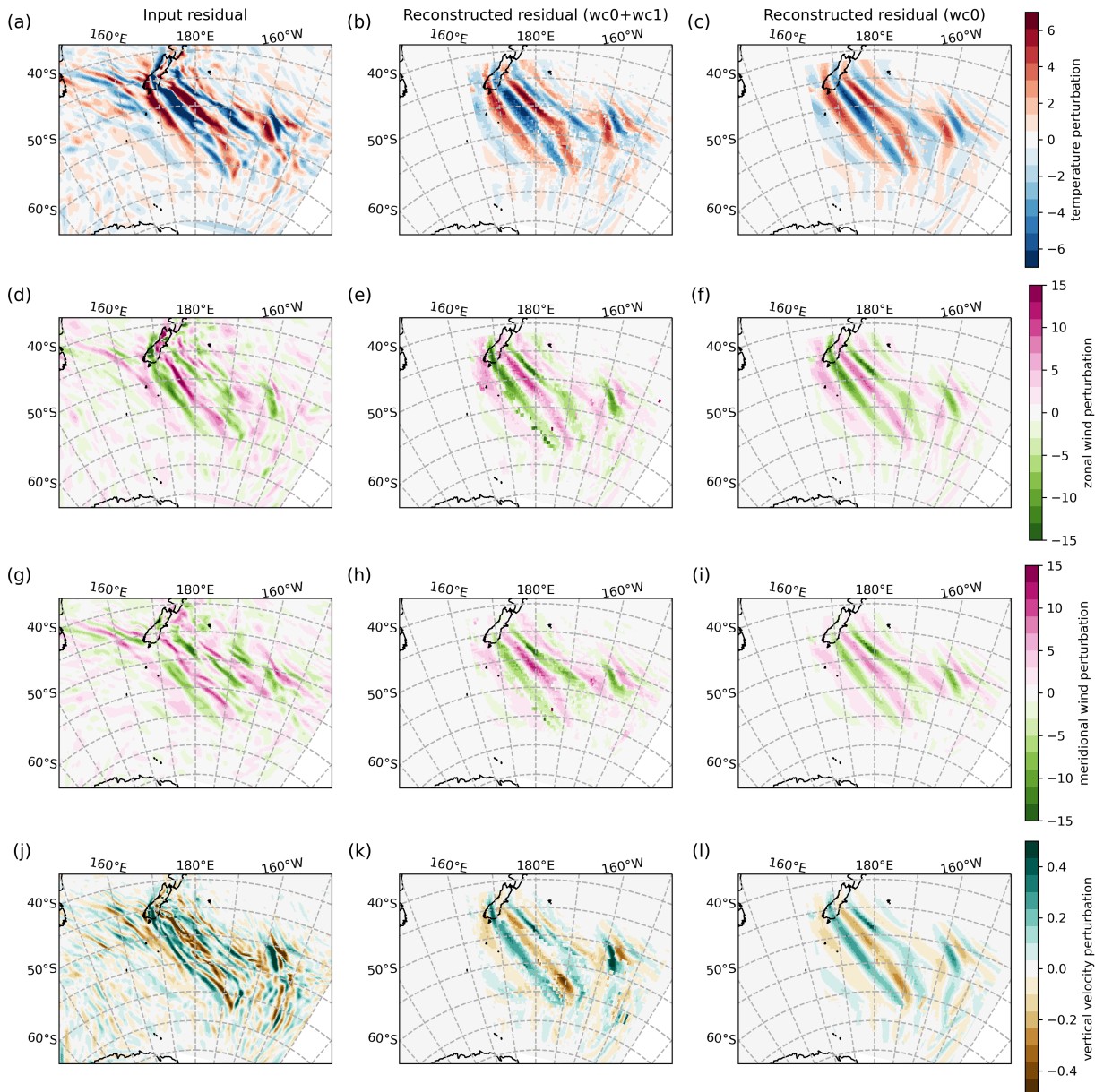

**Figure B1.** Comparison of input and reconstructed perturbation fields of different atmospheric quantities, namely, temperature ((a)-(c)), zonal wind ((d)-(f)), meridional wind ((g)-(i)) and vertical velocity ((j)-(l)) at 25 km altitude. The input perturbation field are shown in the left column. In the middle column, the reconstructed fields are shown that were generated by considering two S3D fitted wave components in the reconstruction from S3D results. The right column shows a reconstruction from S3D results with only one wave component. Please see the text for more details on the generation of those fields.

is closer to the original perturbation with two wave components considered for all considered quantities. Zonal and meridional winds are very similar since the majority of wave perturbations in the case are associated with south-westward pointing wave vectors. Differences between the two horizontal wind components are mainly found where the wave vectors have a preferential southward (patterns more pronounced in meridional wind) or westward (patterns more pronounced in zonal wind) orientation. Differences are found in vertical velocity: The main wave crests and troughs often split into a double-peak structure which shows that waves with shorter horizontal wavelengths are present, which however have the same orientation and location as the larger waves observed in the temperature and horizontal wind perturbations. In general, finer structures are expected in the vertical velocity perturbations than in temperature and horizontal wind perturbations, because the vertical velocity is more sensitive to high frequency waves that on average exhibit shorter horizontal wavelengths (Geller and Gong, 2010).

In general, the reconstruction from the S3D fits from temperature perturbations captures the main features of the vertical velocity wave field though with underestimated peak-to-peak values. This is at least in part due to our assumption of a single sinusoid with constant wave vector throughout the fitting volume as indicated by the reconstruction with two wave components considered. This could be mitigated by refitting the amplitude and phase with a known wave vector in a smaller cube around the cube center (Krisch et al., 2017). It has, however, little effect on the relative distribution and propagation properties of the wave, which is the main aim of this study.

## B2  Results of S3D applied on vertical winds

Analogous to the analysis presented in Section 3.1 for temperature perturbations, we used the S3D method on the corresponding vertical velocity field for the ECMWF operational analysis data of 1 August 2014, 12:00 UTC. Before the fits, the vertical velocity with respect to pressure available in the ECMWF operational analysis dataset was converted into vertical velocity with respect to height assuming hydrostatic conditions. Figure B2 shows the same fields as Figure 3 for the fit from vertical velocity perturbations. Please note that the colormaps in Figure B2 and Figure 3 are chosen to have the same limits for easy comparison, of course with the exception of the amplitude plot in panel (a), which indeed show different quantities.

Both analyses show very similar distributions of the fitted wave parameters, especially in the areas of large temperature amplitudes (regions **I**, **S** and **E**) that we mainly discussed in Sections 3 and 4. The main difference lies in some regions with shorter detected horizontal wavelengths, which can be expected from previous research like Geller and Gong (2010) and seen in the residual fields (Figure B1): vertical velocities emphasize shorter horizontal scales in gravity waves than temperatures. Overall, the momentum flux clusters in the large amplitude regions in the S3D analysis of vertical velocities (Figure B2) exhibit approximately the same locations and similar magnitudes as the results from temperature perturbations.

## B3  Comparison of momentum flux estimates from S3D and WTQ methods

Several methods have been developed to determine gravity wave momentum flux from perturbation fields. The methods vary in spectral and spatial localization and have merit for different scientific questions. Comparisons of different approaches can be found in the methodical papers of Lehmann et al. (2012) and Schoon and Zülicke (2018). In the past, validation of the calculation of momentum flux from S3D has been conducted: The method was compared to Fourier transform of a whole

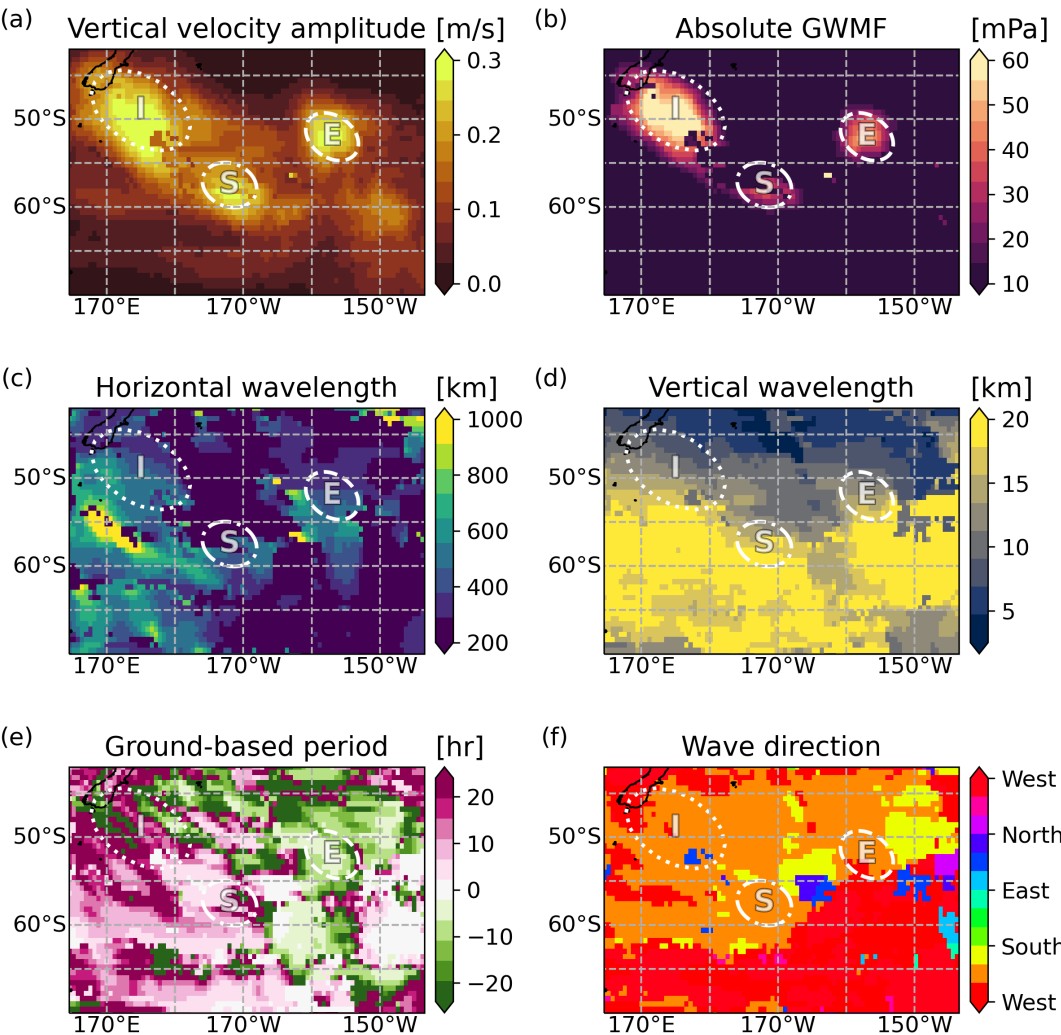

**Figure B2.** Same as Figure 3 but for vertical wind fluctuations

domain in Lehmann et al. (2012); a validation of the S3D method results against products of horizontal wind and vertical velocity amplitudes is included in Preusse et al. (2014); a validation against averaged fluctuations $\overline{u'w'}$ and $\overline{v'w'}$ was shown by Stephan et al. (2019a). The latter cannot take the reduction of pseudo-momentum flux by influence of the Coriolis force into account, but it presents a reliable magnitude estimate.

For the case investigated here, a comparison of momentum flux from S3D with values estimated by the wind and temperature quadratics (WTQ) method (Geller et al., 2013; Stephan et al., 2019a) is shown in Figure B3. Both estimates are averaged over an area spanning from 170° E to 190° E and from 70° S to 40° S for different altitudes, because the WTQ method is only meaningful if averaged over at least a full wave cycle. The gravity wave momentum flux calculated from S3D fits with only

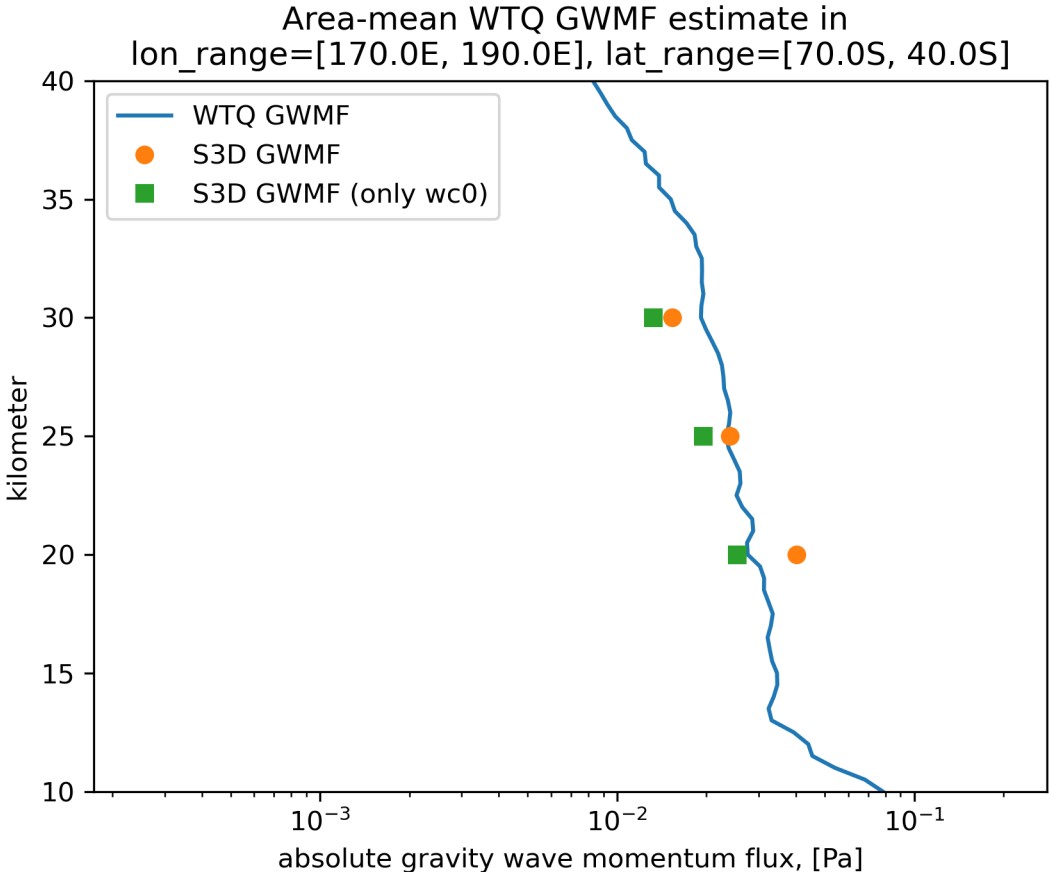

**Figure B3.** Absolute gravity wave momentum flux estimates calculated with two different methods: the blue line shows GWMF from the wind and temperature quadratics (WTQ) approximation used before by Geller et al. (2013) and Stephan et al. (2019a), the orange circles and green squares show GWMF calculated using Eq. 7 from Ern et al. (2004). The green squares only consider one fitted wave component for the calculation, whereas the orange circle show the GWMF for two wave components.

one (the strongest) wave component considered (represented by the green squares) are generally underestimating the WTQ results. WTQ, on the other hand, assumes monochromacy in its derivation. In particular, if vertical velocity and horizontal winds are governed by waves of different wavelengths the method overestimates momentum flux. Given these caveats we find reasonable agreement.

## B4 Influence of parameter choice on the inferred wave origin

As shown in Sections B1 and B2, temperature and horizontal winds emphasise longer horizontal wavelength gravity waves while vertical winds emphasise short horizontal wavelength gravity waves. Thus, we would expect the largest influence on

the raytracing results in using wave characterisation from vertical velocity perturbations instead of temperature perturbations. We conducted an analogous experiment as shown in Section 3.3 with S3D fits from vertical velocities and compared the source regions found in both cases. The equivalent plot to Figure 4 is presented in Figure B4. The inferred source regions remain largely the same, but the weighting of momentum flux contribution among the categories is shifting slightly to 40% for mountain waves, 25% for high-terminated and 34% for low reaching rays compared to 31%, 28% and 42%, respectively, for the temperature-based analysis. This shift from low-reaching to mountain category is to be expected as orography also excites many short-wavelength waves carrying a hgh amount of momentum flux while low-reaching rays from non-orographic sources are mainly associated with long horizontal wavelengths.

In summary, the largest influence on the results is by the reduction of fluctuations to one monochromatic wave per volume. In principle, S3D can decompose in more spectral components, but we decided for this reduction in order to avoid over-interpretation of fitting imperfections. Overall, we find that fluctuations among the different variables are consistent and that the choice of the variable has only minor influence on our main findings.

*Author contributions.* CS, PP, and ME contributed in developing the S3D analysis code. PP, ME and MR supported the development scientifically. CS performed the technical analysis using S3D, the run of the GROGRAT model and developed the ray classification; PP supervised the research and provided scientific support for the full analysis. All coauthors contributed in the interpretation of results, preparation of the paper text and revision of the paper figures.

*Competing interests.* The authors declare that they have no conflict of interest.

*Financial support.* This work was partly funded by Deutsche Forschungsgemeinschaft (DFG) project PR 919/4-2 (MS-GWaves/SV), which is part of the DFG researchers group FOR 1898 (MS-GWaves) and by the German Ministry for Education and Research under grant 01 LG 1907 (project WASCLIM) in the frame of the Role of the Middle Atmosphere in Climate (ROMIC)-program.

*Disclaimer.* TEXT

*Acknowledgements.* The authors would like to sincerely thank Benedikt Ehard for discussion on the horizontal propagation direction of mountain waves, Lars Hoffmann for discussions on the occurence of gravity waves in AIRS observations and Jörn Ungermann for editorial support on the paper text and figures. The authors also gratefully acknowledge Stephen D. Eckermann and Crispin J. Marks for providing the original GROGRAT code. The European Centre for Medium-Range Weather Forecasts (ECMWF) is acknowledged for meteorological data

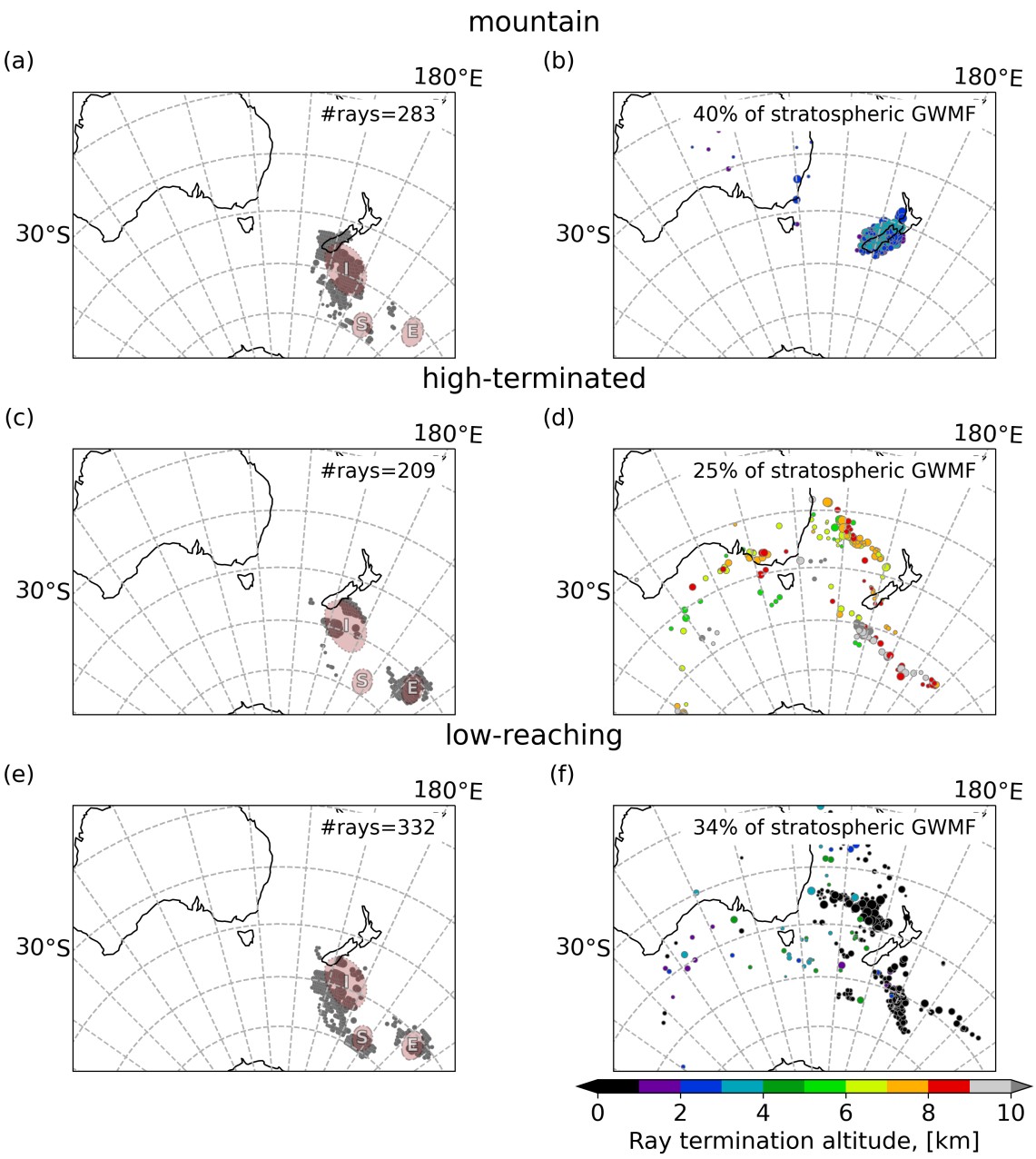

**Figure B4.** Same as Figure 4, but launched from S3D fits of vertical velocity instead of temperature perturbations.

support. Furthermore, ME would like to acknowledge discussions at the International Space Science Institute (ISSI), Bern, with members of the ISSI team "New Quantitative Constraints on Orographic Gravity Wave Stress and Drag".

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
