# Peer review of "Propagation Paths and Source Distributions of Resolved Gravity Waves in ECMWF-IFS analysis fields around the Southern Polar Night Jet"

_Atmospheric Chemistry and Physics, 2021_

## Referee Comment (RC1)

**Review of:** "Propagation paths and source distributions of resolved gravity waves in ECMWF-IFS analysis fields around the southern polar night jet"

by C. Strube et al.

**Recommendation:** Minor revision

This is an interesting and well-written paper that illustrates by means of ray tracing that gravity waves excited in the troposphere over SE Australia and New Zealand can propagate over 1000 km in the meridional direction by the time they reach the middle stratosphere. The results lend support to the idea that GW meridional propagation can explain the "forcing gap" at high southern latitudes that has been implicated in the cold-pole problem of current GCMs and climate models.

Publication is recommended after the authors address the specific comments listed below.

**Specific Comments** (line number):

(19) "to account for lateral propagation": More precisely, to account for *meridional* propagation, which is the major deficiency in current column-based GW parameterizations.

(48) "gravity waves propagate along their phase lines": It would be clearer to write that the *group* propagation of gravity waves occurs along their phase lines.

(257) "Figure 3": Please state explicitly here that these maps are for an altitude of 25 km. This information is given in the figure caption, but it is important enough to include also in the text.

(294) "clear source attribution … and hence can be described by … a single monochromatic wave": I am not sure how it follows that a well-defined source implies a monochromatic wave. For example, the horizontal wavenumber spectrum of GW waves excited by wind blowing over a Gaussian obstacle will itself be a Gaussian, not a monochromatic wave. Perhaps I am misunderstanding what the authors are saying here?

(331) "group velocity falls below a specified threshold": What is the threshold?

(339) "The ray launch and termination points … in Figure 4": Please state explicitly here that the ray launch points (left column of Fig 4) are all at 25 km. The figure caption states this, but this is important information that should be mentioned in the text here, when the reader first encounters Fig. 4.

(371) "The grey dots … are scaled": It might be clearer to write that the size of the gray dots is proportional to the GW momentum flux at the location of each launch point.

(402) "The majority of these waves": More accurately, "the majority of these rays".

(403) "the tropospheric jet upwind of New Zealand": It would be useful to refer here to Fig. 1, where the synoptic situation is shown.

(415) "'mountain' rays travel short [horizontal] distances": Is this because the ratio of horizontal to vertical group velocities $c_{gx}/c_{gz} \sim [k(U-c)]^{-1}$, where $U$ is the background wind, $c \sim 0$, and $U$ is strong in the generation region of the mountain waves and at higher altitudes as they propagate upward?

(453) "contradiction between": It would be useful to explain here, in a sentence or so, what the contradiction is.

(465) "wind and intrinsic phase speed cancel each other": I am not sure what this means. The intrinsic phase speed is $(U-c)$, where $c$ is the ground-based phase speed, so I do not understand in what sense the wind and the intrinsic phase speed "cancel each other". Do you mean to say that certain values of $c$ can reduce the intrinsic phase speed $(U-c)$? Note also that, for mountain waves, $c \sim 0$, so it is not clear how much effect $c$ can have on the intrinsic phase speed.

(466) "the wave drifts with the wind": What does this mean?

(482) "Figure 8 … relative propagation direction": How is the relative angle measured? Clockwise or counterclockwise from the direction of the wind? This needs to be noted here.

(501) "Here 0° represents a direction to the East …":  Note that only the left and middle column show directions with respect to the ground. The right column is the direction of the wave vector relative to the wind.  That aside, it would be less confusing to state that 0° points East, 90° points North, and so on; and that the plots in the first two columns show the direction of the wave vector and the direction of air flow, respectively. Finally, the right column of the figure shows the angle between the wave vector and the wind direction, measured counter-clockwise. The latter should be noted explicitly in the text.

(522) "all directions … are present (panel i)": The spread of angles with respect to the wind in Fig. 9i is somewhat broader than in the other cases, especially that of "mountain" waves (Fig. 9c), but it does not appear to me qualitatively different. In particular, it is not the case that in Fig. 9i "all directions relative to the wind are present". There are virtually no rays with directions between 270° and 90° relative to the wind in Fig. 9i.

**Grammar, syntax, typos:**

(371) "The grey dots, that are marking the launch locations, …" → The grey dots that mark the launch locations… (no commas, since this is a "restrictive" relative clause)

(381) "non" → none

(383) "Australia, however, …" → Australia; however, …

(425) "large amount … of rays travels" → large number … of rays travel

(532) "for at 8 km altitude" → at 8 km altitude

(533) "aligns form the Australian East coast" → extends SE from the East coast of Australia

(538) "for the stratosphere most relevant" → the waves most relevant for the stratosphere

(Fig. 11 caption) "direction where the wave vector is pointing to" → direction of the wave vector

---

## Author Comment (AC1)

**Author's response**

July 20, 2021

**To the editor**

We thank all reviewers for attentively reading our manuscript and their constructive comments. In this revision, we included small changes to address the questions and comments of Reviewer 1's largely favourable review. Reviewer 2 asked about the dependency of our study on the choice of the analysed variable (i.e. temperature) and the consistency between different variables. We have carried out additional analysis to that effect and show the results in the Appendix B and a new subsection 4.5 in the main text. Reviewer 3 has made the suggestion to investigate the dependence of propagation distance on horizontal wavelength which is along with our motivation in the introduction. For this we included a new Figure (now Figure 9). However, we do not agree with the comment of Reviewer 3 about mistakes in Appendix A. Appendix A is in line with the common understanding in the literature (cf. for example the review paper by Fritts and Alexander (2003) and previous work e.g. by Sato and coauthors).

**Reply to Reviewer 1**

We thank the reviewer for her / his very favourable review of the paper and the helpful comments for increasing readability towards more precise and lucid descriptions.

**(19) "to account for lateral propagation": More precisely, to account for meridional propagation, which is the major deficiency in current column-based GW parameterizations.**
changed as suggested

**(48) "gravity waves propagate along their phase lines": It would be clearer to write that the group propagation of gravity waves occurs along their phase lines.**
changed as suggested

**(257) "Figure 3": Please state explicitly here that these maps are for an altitude of 25 km. This information is given in the figure caption, but it is important enough to include also in the text.**
altitude included in text, also in reply to Reviewer 2 a short motivation for the selection of the main analysis altitude is provided.

**(294) "clear source attribution ... and hence can be described by ... a single monochromatic wave": I am not sure how it follows that a well-defined source implies a monochromatic wave. For example, the horizontal wavenumber spectrum of GW waves excited by wind blowing over a Gaussian obstacle will itself be a Gaussian, not a monochromatic wave. Perhaps I am misunderstanding what the authors are saying here?**
In the example of the Gaussian mountain, the horizontal wavenumber spectrum would have a Gaussian amplitude envelope around a main carrying frequency determined by e.g. the width of a mountain ridge for horizontal scales. A dominant vertical wavelength is given by the dispersion relation for mountain waves. Likewise other source processes have dominant scales. In addition, propagation to the analysis altitude tends to separate different scales. Accordingly, in our experience the monochromatic approach is quite successful. The text is reformulated to:

*In addition, gravity waves with amplitudes above a general background level can be expected to have a clear source attribution. Various sources excite gravity waves of scales typical for the particular source process.*

*Even if a source emits a spectrum of waves, propagation to the altitudes we consider will spatially separate the spectral components. Accordingly, it has been shown by Lehmann et al. (2012) for the case of a typhoon that the spectral distribution obtained with a spectral analysis method applied on a larger region around a source is well described by the spectral distribution gained from the monochromatic waves in this region. This facilitates backward ray-tracing to these sources which has been applied in a number of previous studies (Preusse et al., 2014; Krisch et al., 2017; Perrett et al., 2021).*

**(331) "group velocity falls below a specified threshold": What is the threshold?**
0.01 m/s ; inserted in the text

**(339) "The ray launch and termination points . . . in Figure 4": Please state explicitly here that the ray launch points (left column of Fig 4) are all at 25 km. The figure caption states this, but this is important information that should be mentioned in the text here, when the reader first encounters Fig. 4.**
We included:
*All rays were launched from an altitude of 25 km altitude. The altitude of the termination points differs for each ray and is shown by the colour of the location dot in the termination point plots.*

**(371) "The grey dots . . . are scaled": It might be clearer to write that the size of the gray dots is proportional to the GW momentum flux at the location of each launch point.**
Reformulated to
*The grey dots mark the launch locations and their area is chosen proportional to the launch GWMF inferred for the individual ray.*

**(402) "The majority of these waves": More accurately, "the majority of these rays".**
corrected
**(403) "the tropospheric jet upwind of New Zealand": It would be useful to refer here to Fig. 1, where the synoptic situation is shown.**
reference included

**(415) "'mountain' rays travel short [horizontal] distances": Is this because the ratio of horizontal to vertical group velocities cgx/cgz    [k(U − c)]-1, where U is the background wind, c 0, and U is strong in the generation region of the mountain waves and at higher altitudes as they propagate upward?**
For the waves with small angle between wind and wave vector, it is because of the compensation between wind and intrinsic horizontal group velocity. Also really high angles do not occur, which is likely an effect of excitation efficiency. However, at this point of the text the statement is just an observation, and interpretation is included later, so we do not modify the text.

**(453) "contradiction between": It would be useful to explain here, in a sentence or so, what the contradiction is.**
The sentences now read
*In the introduction, we broad an apparent contradiction forward between the modelling study of Holt et al. (2017), who see momentum flux maxima mainly at mid-latitudes for 15 km, and the superpressure balloon observations of Hertzog et al. (2008) and Jewtoukoff et al. (2015), who observe momentum flux maxima associated with the winter polar vortex at 18 km altitude. It seems unlikely that all this momentum is transported laterally over this very small altitude range of only 3 km. However, if waves that remain very close to their sources and have little relevance for the stratosphere were dominant at 15 km altitude but disappear above, that would explain this apparent contradiction in the location of the GWMF maxima.*

**(465) "wind and intrinsic phase speed cancel each other": I am not sure what this means. The intrinsic phase speed is (U − c), where c is the ground-based phase speed, so I do not understand in what sense the wind and the intrinsic phase speed "cancel each other". Do you mean to say that certain values of c can reduce the intrinsic phase speed (U − c)? Note also that, for mountain waves, c    0, so it is not clear how much effect c can have on the intrinsic phase speed.**
Has been changed to

*If they are directed strictly opposite, wind and intrinsic phase speed are of the same magnitude but opposite to each other $(U - \hat{c}) = c_{gb} \approx 0$) and thus the intrinsic group velocity compensates the advection of the wave parcel by the wind and keeps the wave above the source.*

**(466) "the wave drifts with the wind": What does this mean?**
see above

**(482) "Figure 8 ... relative propagation direction": How is the relative angle measured? Clockwise or counterclockwise from the direction of the wind? This needs to be noted here.**
Added clarification to the text.
*$180°$ indicates opposite orientation, the difference angle is counter-clockwise with respect to the wind direction.*

**(501) "Here 0° represents a direction to the East ...": Note that only the left and middle column show directions with respect to the ground. The right column is the direction of the wave vector relative to the wind. That aside, it would be less confusing to state that 0° points East, 90° points North, and so on; and that the plots in the first two columns show the direction of the wave vector and the direction of air flow, respectively. Finally, the right column of the figure shows the angle between the wave vector and the wind direction, measured counter-clockwise. The latter should be noted explicitly in the text.**
Changed as suggested

**(522) "all directions ... are present (panel i)": The spread of angles with respect to the wind in Fig. 9i is somewhat broader than in the other cases, especially that of "mountain" waves (Fig. 9c), but it does not appear to me qualitatively different. In particular, it is not the case that in Fig. 9i "all directions relative to the wind are present". There are virtually no rays with directions between 270° and 90° relative to the wind in Fig. 9i.**
This is to some degree an effect of the plot. The lower altitudes appear as less important simply because the same angle differences correspond to less space on the paper, but actually low-altitude angles are as meaningful as at higher altitudes. This is certainly a disadvantage of this type of plot, but it has many other advantages. In panel (i) around $10\,\mathrm{km}$ and below, angles of very close to 270° are populated which is not the case in the two other groups and definitely not in the 'mountain' rays. We have modified the text to draw more attention to this:
*The direction relative to the wind is generally opposite, that is only the left-hand side of the circle is filled, but there almost all directions occur. In particular, for lower altitudes near $10\,\mathrm{km}$, angles can be very close to $270°$, which discerns the high-terminated rays from the other two groups and in particular from the mountain waves.*

**Reply to Reviewer 2**

We thank Reviewer 2 for her / his helpful comments. We agree that using also the other model quantities for comparison can give a feeling for the consistency of the analysis and thus raise confidence in the main results. However since this is not part of the main message of the paper, as the reviewer pointed out as well, we have added a new appendix performing such consistency checks. The suggestions made will help to improve the quality of the paper.

**1. The investigation is carried out with the identification of gravity wave packets as a starting point. This identification focuses on temperature. This makes sense and is related to the relevance of this variable and the corresponding diagnostic tools in observations. However, since the analysis is carried out on model data, other variables could be used. The vertical velocity in particular is available, and variations will naturally be dominated by gravity waves (no or much less need to remove background large-scale gradients). What are the results of the S3D method on the vertical velocity field (or on other fields, e.g. meridional wind, ageostrophic wind, divergence..)? What is the sensitivity of the method to the choice of the reference variable for the identification of wave packets and what does this imply for the overall interpretation?**
We have added a section to the appendix in the revised version of the paper which tests the results in terms of consistency between temperature, horizontal winds and vertical velocities. One advantage of temperature

and vertical velocities compared to horizontal winds is that the strength of the patterns is independent of the propagation direction. Vertical velocities tend to emphasise shorter scales which leads to somewhat higher momentum fluxes and a stronger emphasis on mountain waves, but does not impact the main conclusions of this study.

**2. The gravity wave momentum flux (GWMF) is calculated, it seems, from the wave characteristics ('GWMF depends both on the squared temperature amplitude and the ratio of horizontal to vertical wavelengths', line 264). Yet from the model output there is much more information available, allowing a direct calculation of the momentum fluxes from the different variables involved, including velocity components. Although such validation is not the focus of the manuscript, a comparison of the two estimates would bring much more confidence to the estimates of GWMF, which is a central quantity in the study of gravity waves. Reducing the uncertainty on the estimate of these fluxes is important and worthwhile.**

There is no gold standard for inferring momentum flux of model-resolved gravity waves and each method has its own advantages and disadvantages. In the appendix we show the consistency of variables using the polarisation relations which is also a test for the wave parameters inferred, thus supports our estimate of GWMF. In addition, we compare with the WTQ method of Geller et al. (2013) and find reasonable agreement. The largest deviations are likely due to the monochromacy assumption, which leads to an underestimation for S3D and an overestimation for WTQ.

**3. Although the tool used for ray-tracing does not require validation, as it has been used and its relevance demonstrated many times, it could be useful to explore a bit the sensitivity of the analysis to some of the methodological choices: if the analysis of the waves is carried out at 30 km instead of 25 km, or at 20 km, are the results essentially unchanged? How well do rays launched from 30 km correspond to the wavefield present at 20 km when they have propagated down to 20 km (with the reserve in mind that part of the waves present at 20 km do not propagate up to 30 km)? Of course, the questions suggested here would require more work than is reasonable, but it would be useful to give some indications for the sensitivity of the analysis to different choices that need to be made in the process.**

This is a very interesting comment.

From experience we had chosen 25 km altitude as an optimum height for the following reasons:

- Most of the lateral propagation already took place at altitudes below, so we can address our question with backward ray-tracing alone while lower altitudes would require backward and forward ray-tracing.

- At 25 km altitude separation of gravity wave variations and planetary-scale background works reliably (Strube et al., 2020), but is more difficult for altitudes of 20 km and below.

- For higher altitudes uncertainties grow because of the longer trajectories.

Accordingly, we have inserted a few sentences motivating the chosen altitude to section 3.1 in which we describe the S3D method and the parameters used for it.

However, performing the same analysis for 20 km height and 30 km height we find significant changes which are consistent with the expected filtering between these altitudes. As these can be best understood after full interpretation of the 25 km results we have introduced a new subsection *4.5 Dependence on the investigation altitude*.

We believe that this adds a deeper understanding of the gravity wave filtering process between the various altitudes (the backward ray-tracing shows only the result of the filtering). Again, we thank the reviewer for making this suggestion!

**4. The origin of the non-orographic waves deserves some more discussion. Although it is quite acceptable that "A detailed discussion of the source processes is however beyond the scope of this study," (l359) some indications of what the sources could be, and some references, should be included. In particular, some information may be present in the output of the ECMWF-IFS. In addition to the vertical velocity plotted in figure 11, the divergence of the horizontal wind should be plotted. It highlights a different part of the spectrum relative to vertical velocity, and is particularly relevant for non-orographic waves, which often have low intrinsic frequencies.**

[Figure]

Figure 1: Caption

As suggested, we have also generated the same plot as Figure 11 for wind divergence. Similar to vertical winds this also shows large-scale, relatively weak features almost parallel to the isobars (i.e. to the horizontal winds). Hence, the divergence confirms our finding that not necessarily the strongest features but those which have a favourable direction for lateral propagation at excitation can make their way to the stratosphere.

**Minor points / Technical corrections**

All technical corrections were implemented as suggested. Implementation of minor points is discussed below.

**l49-50: this is slightly misleading because the distances over which horizontal propagation will matter are much larger than those over which vertical propagation matters. The propagation is in line with the aspect ratio of the wave.**
*For many waves then the ratio of horizontal and vertical wavelengths is comparable to the ratio of vertical to horizontal grid spacing in general circulation models, so that implementing vertical propagation but neglecting horizontal propagation across grid cells is a strong simplification. The exception for which lateral propagation is much less important are those mountain waves ...*

**Reply to Reviewer 3**

**Paramount importance: In line 496 ff you write about the "paramount importance of the relative wave direction for the distance..." in contrast to the wave-refraction effects. Do you mean those waves in Fig. 8 which travel more than 4000 km which are opposite to the wind? This needs to be elaborated and made more pronounced.**
The sentence summarises the paragraphs above. In these paragraphs we will introduce explicit references to the respective panels of Figure 8. Figure 8 shows a clear dependency of the propagation distance (we will exchange the word "paramount") for waves between 0 and 3000km propagation distance from the source. Figure 8 shows a clear correlation, but does not explain the reason for the direction. We are asking the question, whether the waves are oriented favourably already at source or is the direction change by refraction more important. We have reformulated this paragraph:
*The results shown so far in this section suggest that the relative wave direction with respect to the wind is a governing factor for the lateral propagation distance of the low phase-speed gravity waves considered here. The question then is: what determines this angle? In previous studies, horizontal refraction was presented as*

*a major factor for lateral propagation and focusing of gravity waves into the stratospheric jet stream (Sato et al., 2009; Preusse et al., 2009). On the other hand, the sources may generate a favourable direction of the gravity waves from excitation. Hence, what is more important, the direction at excitation or the change of direction due to refraction?*

**Long waves: You show long waves travel long - as may be verified with the ratio of horizontal and vertical group velocities. Jiang et al. (2019) show similar effects as well as Ehard et al. (2017). This needs to be included in the discussion. Are these long-travelling waves the "paramount" waves oriented against the wind?**

In principle, you would expect this. However, please note that this refers to the intrinsic and not the ground-based horizontal group velocities. Thus, a long mountain wave oriented perfectly against the wind does not propagate far, while a much shorter mountain wave at favourable angle could travel much further. We agree, however, with the reviewer that this is a point worth of investigation and have included another figure (Figure 9 in the revision) plus discussion. The results are mixed: there is a maximum of short horizontal wavelength mountain waves - otherwise we do not find a general correlation. Thus, the reason is more likely the details in excitation than the phase-front tilt and could be specific for New Zealand or a more general feature where mountain wave excitation is most effective for such ridges perpendicular to the wind.

**Mountain-wave directions: In the appendix you show that ground-based (apparent) group velocity is perpendicular to the wavenumber vector respectively parallel to the phase lines (or the mountain ridge). Can this detail be used to interpret the wave propagation? What was your intention to show Fig. A1 when you do not refer to it in the main text?**

The main take-home message of this figure is that ground-based horizontal group velocity is parallel to the phase fronts. We have used this fact, for instance, in section 2.3 in order to identify directly from the temperature perturbations which part of the wave field is potentially due to trailing mountain waves from New Zealand and which parts of the wave field likely have different sources. At this point we referenced the appendix, but will now include also the reference to the figure.

The second take-home message is: Waves at larger angles to the winds propagate further. This is important for all the angle discussions. We have added a further reference to the appendix, there.

**225 Degree: What is the reason for the major wave orientation seen in Fig. 9? It is not apparent in Jiang et al. (2019), but is also present in Ehard et al. (2017). They argue it is a refraction effect. Please, discuss this issue.**

This is the point of Figure 9, in particular of the left column. The modified introductory sentences should (see above) clarify this. If the distribution forms a straight line from origin to outer diameter at the same angle that means refraction can be neglected (most of mountain and low-reaching rays). Where that forms a curved shape, refraction is important (high-terminated).

**Appendix: This part of the paper has to be rewritten because it contains a number of mistakes. For example, equation (A1) is wrong because phase velocity is the ratio of frequency and wavenumber and it is NOT parallel to the wavenumber vector ( k, l ). In the text, you write in line 595 that the phase and group veocity are equal but you do not give a dispersion relation. For mid-frequency waves with**
**it is definitely NOT. While eq. (A2,A3) is formally correct for such waves, it might be easily extended for arbitrary wavenumbers ( k, l ) and winds ( u, v ). Also figure A1 needs revision: there should be intrinsic (flow-relative) group velocity instead of "horizontal phase speed vector" because the vector addition is fo the group velocity. Further, I suggest to extend the consideration with a decomposition of the group velocity in a component parallel and perpendicular to the wavenumber.**

In reply to this review we quote Fritts and Alexander (2003) from their review paper:
*Note that the phase speed is not a vector quantity, although wave phase propagation has a direction given by the vector $(k, l, m)$.*

This is the 3D propagation of the phase lines as indicated for instance by Andrews et al. (1987) in their Figure 4.19. For critical level considerations the horizontal phase speed is commonly used which is then defined as the phase speed in the direction of the horizontal wave vector and thus amounts to

$$|\hat{c}_h| = \frac{\hat{\omega}}{|\boldsymbol{k_h}|} = \frac{\hat{\omega}}{\sqrt{k^2 + l^2}} = \frac{N}{|m|} \tag{1}$$

where the right hand side is based on the mid-frequency dispersion relation

$$\hat{\omega}^2 = \frac{N^2 k_h^2}{m^2} \tag{2}$$

(Please note that we used the notations introduced in the paper here.)

This definition of the horizontal phase velocity is in accordance with textbooks (Andrews et al., 1987), equation 33 of Fritts and Alexander (2003) and for instance blocking diagrams as constructed by **?**. It is also the basis of many gravity wave parametrization schemes which usually choose a 2D formulation with main horizontal axis in the direction of the horizontal wave vector for simplification.

The group velocity, in contrast, *is* a vector and defined as

$$\hat{c}_g = \left( \frac{\partial \hat{\omega}}{\partial k}, \frac{\partial \hat{\omega}}{\partial l}, \frac{\partial \hat{\omega}}{\partial m} \right) \tag{3}$$

We can now apply the derivative, for instance for $k$, on both the left and right hand side of the dispersion relation

$$\frac{\partial \hat{\omega}^2}{\partial k} = 2\hat{\omega} \frac{\partial \hat{\omega}}{\partial k}; \quad \frac{\partial \hat{\omega}^2}{\partial k} = 2 \frac{N^2 k}{m^2} = 2 \frac{N^2 k_h^2}{m^2} \frac{k}{k_h^2} \Rightarrow \frac{\partial \hat{\omega}}{\partial k} = \frac{\hat{\omega} k}{k_h^2} \tag{4}$$

Calculating the derivatives for $l$ and $m$ analogously we thus gain

$$\hat{c}_g = \left( \frac{\hat{\omega}}{k_h} \frac{k}{k_h}, \frac{\hat{\omega}}{k_h} \frac{l}{k_h}, -\frac{\hat{\omega}}{m} \right) \tag{5}$$

Which leads finally to the equality between horizontal phase speed and horizontal group velocity in accordance with the picture of Andrews et al. (1987). One can do the analogous calculation for low frequency waves and finds the same equality of horizontal phase speed and group velocity.

Furthermore, our Figure is in good agreement to the one presented by Sato et al. (2012), only that we emphasise the orientation of the horizontal phase fronts as we directly interpret structures in temperature residuals.

The choice of the direction of u was made without loss of generality (similar as to the choice of Andrews et al. (1987) to place the horizontal wavevector in the x direction and of Sato et al. (2012)) and the inferred behaviour is still valid for other choices of the wind direction. Inferring this with arbitrary wind directions would not add to the take-home message but substantially increase the mathematical complexity and we hence prefer the simpler equations.

We have included the definition of horizontal phase speed with reference to Fritts and Alexander (2003) and provided the mid-frequency version of the dispersion relation in the text.

**References**

D. G. Andrews, J. R. Holton, and C. B. Leovy. *Middle Atmosphere Dynamics*, volume 40 of *International Geophysics Series*. Academic Press, 1987.

D. Fritts and M. Alexander. Gravity wave dynamics and effects in the middle atmosphere. *Rev. Geophys.*, 41(1), APR 16 2003. ISSN 8755-1209. doi: 10.1029/2001RG000106.

M. A. Geller, M. J. Alexander, P. T. Love, J. Bacmeister, M. Ern, A. Hertzog, E. Manzini, P. Preusse, K. Sato, A. A. Scaife, and T. Zhou. A comparison between gravity wave momentum fluxes in observations and climate models. *J. Clim.*, 26(17):6383–6405, SEP 2013. ISSN 0894-8755. doi: 10.1175/JCLI-D-12-00545.1.

A. Hertzog, G. Boccara, R. A. Vincent, F. Vial, and P. Cocquerez. Estimation of gravity wave momentum flux and phase speeds from quasi-Lagrangian stratospheric balloon flights. part ii: Results from the vorcore campaign in antarctica. *J. Atmos. Sci.*, 65(10):3056–3070, 2008. doi: 10.1175/2008JAS2710.1.

L. A. Holt, M. J. Alexander, L. Coy, C. Liu, A. Molod, W. Putman, and S. Pawson. An evaluation of gravity waves and gravity wave sources in the Southern Hemisphere in a 7 km global climate simulation. *Quart. J. Roy. Meteorol. Soc.*, 143(707, B):2481–2495, JUL 2017. ISSN 0035-9009. doi: 10.1002/qj.3101.

V. Jewtoukoff, A. Hertzog, R. Plougonven, A. de la Camara, and F. Lott. Comparison of Gravity Waves in the Southern Hemisphere Derived from Balloon Observations and the ECMWF Analyses. *J. Atmos. Sci.*, 72(9):3449–3468, SEP 2015. ISSN 0022-4928. doi: 10.1175/JAS-D-14-0324.1.

I. Krisch, P. Preusse, J. Ungermann, A. Dörnbrack, S. D. Eckermann, M. Ern, F. Friedl-Vallon, M. Kaufmann, H. Oelhaf, M. Rapp, C. Strube, and M. Riese. First tomographic observations of gravity waves by the infrared limb imager GLORIA. *Atmos. Chem. Phys.*, 17(24):14937–14953, 2017. doi: 10.5194/acp-17-14937-2017.

C. I. Lehmann, Y.-H. Kim, P. Preusse, H.-Y. Chun, M. Ern, and S.-Y. Kim. Consistency between fourier transform and small-volume few-wave decomposition for spectral and spatial variability of gravity waves above a typhoon. *Atmos. Meas. Tech.*, 5(7):1637–1651, 2012. doi: 10.5194/amt-5-1637-2012.

J. A. Perrett, C. J. Wright, N. P. Hindley, L. Hoffmann, N. J. Mitchell, P. Preusse, C. Strube, and S. D. Eckermann. Determining gravity wave sources and propagation in the southern hemisphere by ray-tracing AIRS measurements. *Geophys. Res. Lett.*, 48(2):e2020GL088621, JAN 28 2021. ISSN 0094-8276. doi: 10.1029/2020GL088621.

P. Preusse, S. D. Eckermann, M. Ern, J. Oberheide, R. H. Picard, R. G. Roble, M. Riese, J. M. Russell III, and M. G. Mlynczak. Global ray tracing simulations of the SABER gravity wave climatology. *J. Geophys. Res. Atmos.*, 114, 2009. doi: 10.1029/2008JD011214.

P. Preusse, M. Ern, P. Bechtold, S. D. Eckermann, S. Kalisch, Q. T. Trinh, and M. Riese. Characteristics of gravity waves resolved by ECMWF. *Atmos. Chem. Phys.*, 14(19):10483–10508, 2014. doi: 10.5194/acp-14-10483-2014.

K. Sato, S. Watanabe, Y. Kawatani, Y. Tomikawa, K. Miyazaki, and M. Takahashi. On the origins of mesospheric gravity waves. *Geophys. Res. Lett.*, 36, OCT 7 2009. ISSN 0094-8276. doi: 10.1029/2009GL039908.

K. Sato, S. Tateno, S. Watanabe, and Y. Kawatani. Gravity wave characteristics in the southern hemisphere revealed by a high-resolution middle-atmosphere general circulation model. *J. Atmos. Sci.*, 69(4):1378–1396, 2012. doi: 10.1175/JAS-D-11-0101.1.

C. Strube, M. Ern, P. Preusse, and M. Riese. Removing spurious inertial instability signals from gravity wave temperature perturbations using spectral filtering methods. *Atmos. Meas. Tech.*, 13(9):4927–4945, 2020. doi: 10.5194/amt-13-4927-2020. URL https://amt.copernicus.org/articles/13/4927/2020/.

---

## Referee Report (RR1)

**Review of:** "Propagation paths and source distributions of resolved gravity waves in ECMWF-IFS analysis fields around the southern polar night jet" (revised)

by C. Strube et al.

Recommendation: Accept with minor revisions

The authors have responded satisfactorily to almost all of my comments. The paper is now acceptable for publication, subject to addressing the comments (plus a few editorial suggestions) listed below. The line numbers refer to the tracked-changes version of the revised paper.

**Specific Comments:**

(304) "Various sources excite gravity waves of scales typical...": This strikes me as opaque. The point is that the wavenumber spectrum generated by a source depends on the spatial scale of the source. How about something like "Sources of gravity waves generate a wavenumber spectrum that depends on the physical dimensions of the source"?

(307) "the spectral distribution": Do you mean the wavenumber spectrum? Spectral distribution could also refer to the frequency spectrum.

(308) "descried [sic] by the spectral distribution gained from the monochromatic waves in this region": I believe you meant "described" instead of "descried". That aside, I do not understand what you mean by "the distribution gained from the monochromatic waves". What monochromatic waves are you referring to?

Note that all of this confusion appears to have arisen following my original comment that any real-world topographic source will excite a spectrum in wavenumber: For the very simple example of a Gaussian source, the wavenumber spectrum is a Gaussian red spectrum, not a spectrum "centered at the main carrying frequency" [sic—I presume you meant wavenumber here]. That is, the Gaussian width, *L*, of an obstacle in physical space,  $h' = \exp[-k^2/(2L^2)]$ , is inversely related to the Gaussian width of its spectrum in wavenumber space, ~  $\exp[-k^2/(2\kappa^2)]$ , with  $\kappa = L^{-1}$ .

Now, I presume all that you mean is that the physical appearance of the wavepacket excited by orography is dominated by wavelength  $\kappa^{-1} \sim L$ , which is fine. But it would be useful to state things more clearly.

(469) "broad an apparent contradiction forward": I think you mean "brought forward an apparent contradiction", or more simply "highlighted an apparent contradiction".

(477) "require a modeling study upwards from relevant sources": It might be clearer to write "require a modeling study of upward propagation from relevant sources".

(488) "wind and intrinsic phase speed are of the same magnitude but opposite to each other  $((U - \hat{c}) = c_{gb} \approx 0)$ ": This does not make sense because  $(U - \hat{c})$  is the intrinsic phase speed. I

think the source of the confusion is in the use of the word "intrinsic". The common use of the term refers to the value relative to the background flow. See for example:

https://glossary.ametsoc.org/wiki/Intrinsic\_wave\_frequency

In any event, for mountain waves the phase speed relative to the ground is (using your symbol)  $\hat{c}$ , and  $\hat{c} \sim 0$  because the waves are generated by wind blowing over fixed topography (a non-steady wind can still generate a spectrum in frequency/phase velocity, but that is another matter). Insofar as  $\hat{c} \sim 0$ , the waves do not propagate with respect to the ground, but do so with respect to the background flow, with intrinsic phase velocity U and intrinsic frequency kU. The slope of the wave ray with respect to the source (for the limiting case of hydrostatic, plane-parallel waves,  $(k/m)^2 << 1$ ) is  $\Delta x/\Delta z = c_{gx}/c_{gz} \sim -m/k$ ; thus, for positive U the ray tilts toward negative x with increasing altitude.

Nothing stated here invalidates any of your conclusions, but it is important to get these explanations straight and set them down clearly.

(488) "intrinsic group velocity": Note that this is redundant. Group velocity is defined in terms of intrinsic frequency, k(U-c).

(561) "that is only" => that is, only

(563) "discerns": I think you mean "distinguishes".

(643) "This would mark a paradigm shift ... from waves that are horizontally refracted ... to waves that feature southward orientation already at source altitude": A minor comment here is that this statement strikes me as overly dramatic. The fact that there may be sources that produce wavetrains with group velocity oriented in a non-zonal direction is hardly a paradigm shift. In some regions, such waves might dominate, whereas in other situations the refraction mechanism might be important. Note, by the way, that in studies such as Sato (2009) refraction into the core of the SH polar night jet occurs over large vertical distances, and the mechanism was proposed to explain forcing at 0.1-1 hPa, i.e., in the upper stratosphere and lower mesosphere.

---

## Author Response (AR2)

**Author's response**

October 14, 2021

**To the editor**

We thank the reviewers again for their constructive comments. In this revision, we addressed the comments of Reviewer 1 asking for clarity in a few statements by reworking our text in the indicated places. We still don't agree with Reviewer 3's comment on correction of our Appendix A. However, to convey our statement more clearly we have revised Fig. A1 and the text in a few places.

**Reply to Reviewer 1**

(304) "Various sources excite gravity waves of scales typical...": This strikes me as opaque. The point is that the wavenumber spectrum generated by a source depends on the spatial scale of the source. How about something like "Sources of gravity waves generate a wavenumber spectrum that depends on the physical dimensions of the source"?

We have replaced the sentence by your recommendation.

(307) "the spectral distribution": Do you mean the wavenumber spectrum? Spectral distribution could also refer to the frequency spectrum.

Yes, we do. We have specified that in the manuscript accordingly.

(308) "descried [sic] by the spectral distribution gained from the monochromatic waves in this region": I believe you meant "described" instead of "descried". That aside, I do not understand what you mean by "the distribution gained from the monochromatic waves". What monochromatic waves are you referring to?

Note that all of this confusion appears to have arisen following my original comment that any real-world topographic source will excite a spectrum in wavenumber: For the very simple example of a Gaussian source, the wavenumber spectrum is a Gaussian red spectrum, not a spectrum "centered at the main carrying frequency" [sic—I presume you meant wavenumber here]. That is, the Gaussian width, L, of an obstacle in physical space, h' = exp[-x2/ (2 L2)], is inversely related to the Gaussian width of its spectrum in wavenumber space, exp[-k2/(2 k2]), with k = L-1. Now, I presume all that you mean is that the physical appearance of the wavepacket excited by orography is dominated by wavelength k -1 L, which is fine. But it would be useful to state things more clearly.

We have reformulated the respective paragraph. The intention here was to state the fact that a few wavenumbers in local fits are usually enough to describe the wave composition. These few waves then allow meaningful propagation studies of localised wave packets.

(469) "broad an apparent contradiction forward": I think you mean "brought forward an apparent contradiction", or more simply "highlighted an apparent contradiction". We have

adapted the text according to your recommendations. This expression is indeed much better!

(477) "require a modeling study upwards from relevant sources": It might be clearer to write "require a modeling study of upward propagation from relevant sources".

Thank you for the suggestion. We have adapted the text accordingly.

(488) "wind and intrinsic phase speed are of the same magnitude but opposite to each other  $((U\hat{c}) = c_{gb} \approx 0)$ ": This does not make sense because  $U - \hat{c}$  is the intrinsic phase speed. I think the source of the confusion is in the use of the word "intrinsic". The common use of the term refers to the value relative to the background flow. See for example: https://glossary.ametsoc.org/wiki/Intrinsic\_wave\_frequency In any event, for mountain waves the phase speed relative to the ground is (using your symbol)  $\hat{c}$ , and  $\hat{c} = 0$  because the waves are generated by wind blowing over fixed topography (a nonsteady wind can still generate a spectrum in frequency/phase velocity, but that is another matter). Insofar as  $\hat{c} = 0$ , the waves do not propagate with respect to the ground, but do so with respect to the background flow, with intrinsic phase velocity U and intrinsic frequency kU. The slope of the wave ray with respect to the source (for the limiting case of hydrostatic, plane-parallel waves,  $(k/m)^2$ ; 1) is Dx/Dz = cgx/cgz - m / k; thus, for positive U the ray tilts toward negative x with increasing altitude. Nothing stated here invalidates any of your conclusions, but it is important to get these explanations straight and set them down clearly.

Thank you for the detailed explanation. We agree with your assessment that our description might have been confusing and tried to clarify it by simplification. The main point we wanted to convey here is the fact, that in the special case of a pure" mountain wave the wave intrinsic phase speed and the background wind form a balance where on compensates the other.

(488) "intrinsic group velocity": Note that this is redundant. Group velocity is defined in terms of intrinsic frequency, k (U - c).

This part was removed in the course of the previous change.

(561) "that is only" =; that is, only

We have added the colon.

(563) "discerns": I think you mean "distinguishes".

We agree and have exchanged the wording.

(643) "This would mark a paradigm shift ... from waves that are horizontally refracted ... to waves that feature southward orientation already at source altitude": A minor comment here is that this statement strikes me as overly dramatic. The fact that there may be sources that produce wavetrains with group velocity oriented in a non-zonal direction is hardly a paradigm shift. In some regions, such waves might dominate, whereas in other situations the refraction mechanism might be important. Note, by the way, that in studies such as Sato (2009) refraction into the core of the SH polar night jet occurs over large vertical distances, and the mechanism was proposed to explain forcing at 0.1-1 hPa, i.e., in the upper stratosphere and lower mesosphere.

We have toned the statement down a little. The final sentence reads now: Our case study therefore suggests to shift the focus of investigation from waves that are horizontally refracted by strong wind gradients into the polar night jet to waves that feature southward orientation already at source altitude, which could be more important also in general.

**Reply to Reviewer 3**

The scheme as presented in the appendix needs correction by substitution of "hor. phase speed" by "intrinsic group velocity". See Fig. 6 of Sato et al. (2012) which is also constructed of group velocities. ...

It is correct that the schematic drawing of Sato et al. 2012 refers to group velocity. Here we want to make the specific point that in mid-frequency approximation a mountain wave propagates along it's phase fronts. This requires the, in the community well known, equality of horizontal phase speed along the horizontal wave vector and horizontal group velocity. The use of a horizontal phase speed in the direction of the horizontal wave vector is one of the basic definitions used in the GW community. The whole fundamental discussion of critical level filtering, which is for instance basis of all GW parametrization schemes, founds on it. We have discussed this in length in our previous reply with several high ranking examples from the literature given. It is, however, true that the equality of horizontal phase speed and horizontal group speed is given only for mid frequency GWs. We will make that more transparent by adding a comment to the figure. The text to the figure now reads:

Schematic view of the horizontal propagation of a mountain wave. When there is an angle between wave vector and wind, the resulting ground-based group velocity is along the phase line. This relation holds strictly for mid-frequency mountain waves, for which horizontal group velocity and horizontal phase speed are equal, and is an approximation for high-frequency mountain waves.